

# Parametrization and thermodynamic scaling of pair correlation functions for the fractional quantum Hall effect

Jørgen Fulsebakke[1], Mikael Fremling[1,2], Niall Moran[1] and J. K. Slingerland[1,3]

**1** Department of Theoretical Physics, National University of Ireland, Maynooth, Ireland
**2** Institute for Theoretical Physics, Center for Extreme Matter and Emergent Phenomena, Utrecht University, Princetonplein 5, 3584 CC Utrecht, The Netherlands
**3** Dublin Institute for Advanced Studies, School of Theoretical Physics, 10 Burlington Rd, Dublin, Ireland

## Abstract

The calculation of pair correlations and density profiles of quasiholes are routine steps in the study of proposed fractional quantum Hall states. Nevertheless, the field has not adopted a standard way to present the results of such calculations in an easily reproducible form. We develop a polynomial expansion that allows for easy quantitative comparison between different candidate wavefunctions, as well as reliable scaling of correlation and quasihole profiles to the thermodynamic limit. We start from the well-known expansion introduced by Girvin [PRB, 30 (1984)] (see also [Girvin, MacDonald and Platzman, PRB, 33 (1986)]), which is physically appealing but, as we demonstrate, numerically unstable. We orthogonalize their basis set to obtain a new basis of modified Jacobi polynomials, whose coefficients can be stably calculated. We then apply our expansion to extract pair correlation expansion coefficients and quasihole profiles in the thermodynamic limit for a wide range of fractional quantum Hall wavefunctions. These include the Laughlin series, composite fermion states with both reverse and direct flux attachment, the Moore-Read Pfaffian state, and BS hierarchy states. The expansion procedure works for both abelian and non-abelian quasiholes, even when the density at the core is not zero. We find that the expansion coefficients for all quantum Hall states considered can be fit remarkably well using a cosine oscillation with exponentially decaying amplitude. The frequency and the decay length are related in an intuitive, but not elementary way to the filling fraction. Different states at the same filling fraction can have distinct values for these parameters. Finally, we also use our scaled correlation functions to calculate estimates for the magneto-roton gaps of the various states.



# 1 Introduction

The fractional quantum Hall effect (FQHE) [1,2] has long been a phenomenon of considerable interest; for its surprising experimental properties, rich theoretical description, and potential usefulness in the field of quantum computing [3]. Among other approaches, the construction of trial wavefunctions has proven highly successful at modeling FQHE systems. The pair correlation function, $g(r)$, giving information about the probability of finding two electrons at a relative distance $r$, constitutes an important tool when examining a trial wavefunction. It can be used to estimate the ground state energy and, through the single-mode approximation,

the gap to neutral excitations [4]. The general shape of the graph of $g(r)$ is also often used in more qualitative arguments to check that the trial wavefunction represents an interacting incompressible quantum liquid. Such incompressibility is characterized by damped oscillatory behavior. Further, a modified shape of the correlation hole is used as an indicator of pairing or clustering behavior, often associated with the presence non-Abelian anyons in the excitation spectrum. Similarly, when wavefunctions for quasihole excitations are available, the density of the system in the presence of a quasihole is an important quantity. The density then reflects the typical size of the quasihole, contains information about quasihole correlations, and allows for electrostatic calculations involving quasiholes. For example, quasiholes of simple FQH states, such as the Laughlin states, tend to have zeros in the electron density near the center of the excitation. In contrast, quasiholes in paired and clustered states may only show a depression of the electron density but no zeroes.

It is straightforward to find an estimate for the pair correlation function of a real-space trial wavefunction involving a finite number of particles $N$ using Monte Carlo sampling. It is natural to represent the results using graphical plots, and this is the most common approach in the literature. See *e.g.* refs. [5–16] for some new and old example plots of pair correlation functions for FQHE trial wavefunctions. See also the works of refs. [17, 18] on pair correlation functions in fractional Chern insulators, where the correlation profiles reflects the FQHE nature of these systems. There are, however, reasons to seek a more quantitative and reproducible form of $g(r)$: Rigour when comparing the pair correlations of different wavefunctions, extrapolation to the thermodynamic limit $N \to \infty$, reusability for the future, and computing the single-mode approximation are just a few reasons. One may also hope that a more quantitative approach could help put some of the intuitive arguments about compressibility etc., which are based on the shape of the graph, on a more rigorous footing.

This paper presents a polynomial expansion of FQH correlation functions and quasiparticle density profiles that allow for easy high-precision reproduction of the complete correlation functions and scaling to the thermodynamic limit. We also demonstrate how the single-mode approximation of the dispersion can be calculated directly from the expansion coefficients. We provide expansion coefficients for the correlations functions and quasihole densities, as well as dispersion results for several prominent quantum Hall states. These should be useful in future studies of those states and comparisons with alternative or newly proposed trial wavefunctions. In particular, we present extrapolations of the pair correlation functions and quasihole profiles to the thermodynamic limit. This is of interest since it is only in this limit that observables are free from finite-size effects, as should be expected in the bulk of the actual physical system; such extrapolations are standard for *i.e.* ground state energy calculations but have so far not been attempted for pair correlation functions.

The expansion we introduce is inspired by an earlier expansion of $g(r)$ developed by Girvin [5], (see also Ref. [4]). While this earlier expansion is physically appealing it is, as we will show, numerically unstable, which makes it unsuitable for quantitative analysis at large sizes and in particular for extrapolation to the thermodynamic limit. The instability is due to high overlaps between Girvin's basis functions. The alternative basis we use is orthogonal and can in fact be obtained from Girvin's basis by direct Gram-Schmidt orthogonalization.

The rest of this paper is organized as follows: In section 2 we review the non-orthogonal pair correlation expansion on the plane laid out in Ref [5], and some of its shortcomings. In section 3 we adapt the non-orthogonal planar expansion to the sphere, and produce a numerically stable expansion by orthogonalizing the basis functions. In section 4 we benchmark the orthogonal expansion against the previous non-orthogonal one and discuss *e.g.* convergence and scaling behavior. In section 5 we apply the orthogonal pair correlation expansion to determine the shape of the correlation function, in the thermodynamic limit, for several prominent FQH states. We also determine the density of abelian and non-abelian quasiholes on top of the

Laughlin and Moore-Read states. Finally, in section 6, as a concrete application, we discuss in some detail how one estimates the magneto-roton collective mode directly from the expansion coefficients obtained in section 5. An early version of this work appeared in the PhD thesis by J.F [19]. The expansion coefficients (including their uncertainties) for all the states treated in this work is available in digital form in the supplementary material.

## 2 The pair correlation function and planar expansion

The general pair correlation function for a system of $N$ electrons with coordinates $\{r_1, r_2, \dots, r_N\}$ is formally defined through the electron wavefunction $\Psi$ as

$$g(r_1, r_2) = \frac{N(N-1)}{\rho^2} \int \prod_{i>2}^{N} dS_i \, |\Psi(r_1, r_2, \dots, r_N)|^2, \tag{1}$$

where $dS_i$ is the surface measure associated with the configuration space of particle $i$ and $\rho$ is the average density. This definition exploits the fact that $g$ does not depend on the chosen pair, due to the permutation symmetry of $|\Psi|$. The function $g(r_1, r_2)$ is proportional to the probability density of finding one particle at $r_1$ and another at $r_2$. It is normalised by $\rho^{-2}$ so that it equals one when the particles' positions are completely uncorrelated. This normalization removes the asymptotic dependence on the density, which in the FQHE setting, is proportional to the filling fraction. This also means that this $g$ is not strictly speaking a probability density.

Assuming isotropy and homogeneity of the system leads to two simplifications. First, the center of mass of the particle pair is irrelevant, and we use this to fix one coordinate at the origin, *i.e.* we set $r_1 = 0$. Secondly $g$ should only depend on the distance $r = |r_{12}| = |r_2 - r_1|$ between the two particles. Using this we define the reduced pair correlation

$$g(r) = \frac{N(N-1)}{\rho^2} \int \prod_{i>2}^{N} dS_i \, |\Psi(0, r_{12}, \dots, r_N)|^2, \tag{2}$$

which will be the focus of this paper.

An approximate plot of the pair correlation can be obtained using the average values $\bar{g}_i$ of $g$ on the intervals $[r_i, r_{i+1}]$ for some chosen set of $r_j$. The bin values are given as

$$\overline{g}_i = \int_{r_i}^{r_{i+1}} g(r) dS_r \Big/ \int_{r_i}^{r_{i+1}} dS_r, \tag{3}$$

with the integration measure $dS_r = 2\pi r \, dr$ on the infinite plane. The approximate values $\bar{g}_i$ can be computed using Monte Carlo methods for real-space wavefunctions $\Psi$.

The lowest Landau level single particle wavefunctions on the disk are given by $\phi_m \propto z^m \exp(-|z|^2/4)$ in terms of the complex position coordinate $z = x + iy$. Girvin used these, together with the independence of $g$ on the center of mass and circular symmetry to construct the following expansion [5]:

$$g(r) = 1 - e^{-r^2/2} + \sum_{m=1,\text{ odd}}^{\infty} c_m h_m(r), \tag{4}$$

$$h_m = \frac{2}{m!} \left( \frac{r^2}{4} \right)^m e^{-r^2/4}, \tag{5}$$

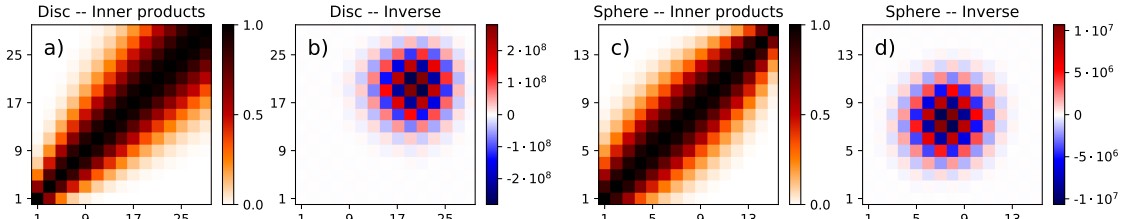

Figure 1: a) The elements of the overlap matrix $M_{mk}$ in (8), for $m, k = 1, 3, \ldots 29$ and b) it's inverse $M_{mk}^{-1}$. All elements of $M_{mk}$ are positive and many off-diagonal elements are of order unity. As a result, the inverse has many elements on the order of $10^8$! Panels c) and d) show the same phenomena for the spherical geometry orbitals $f_k(\eta)$ in eq. 13 and eq. 15, with $2Q = 15$.

where $r = |z|$, and the index $m$ is restricted to odd integers because the electrons are fermions.[1] The coefficients $c_m$ should eventually decrease in size as $m$ increases, so that only a limited number are necessary to represent the correlation function. The first two terms in (4) correspond to the pair correlation of the non-interacting integer quantum Hall state at filling fraction $\nu = 1$ [20],

$$g_1(r) = 1 - e^{-r^2/2}. \tag{6}$$

Thus, we may think of $g_1(r)$ as a "reference correlation function" and the coefficients $c_m$ measure the deviation from this "reference correlation function" of $\nu = 1$. Note that one may in principle use any function instead of $g_1(r)$ as the "reference correlation function", and this will in turn of course affect the expansion coefficients $c_m$.

For future convenience we define an inner product for correlation functions, as

$$\langle f | g \rangle = \int dS_r \, f(r) g(r), \tag{7}$$

where, just as in (3), $dS_r = 2\pi r \, dr$ is the integration measure for $r$. Using this definition of the inner product, Girvin's expansion functions $h_m(r)$ are normalized, $\langle h_m | h_m \rangle = 1$.

The expansion in (4) is intended to make it possible to represent the pair correlation to a high accuracy using only a handful of coefficients. The coefficients themselves, however, are numerically unstable, as expanded upon and demonstrated in section 4. This expansion is therefore unsuitable for applications that use the coefficients directly; *e.g.* comparing coefficients of different wavefunctions or scaling the coefficients with system size.

To get an idea of how the instability in this basis choice comes about, we refer to Fig. 1. There, the overlap matrix

$$M_{mn} = \langle h_m | h_n \rangle = \frac{\left(\frac{m+n}{2}\right)! \left(\frac{m+n}{2}\right)!}{n! \, m!}, \tag{8}$$

and its inverse are plotted for the first 15 terms $m = 1, 3, \ldots, 29$ of (4). As should be evident, all elements of $M$ are positive definite, and therefore the functions $h_k(r)$ are forming a non-orthogonal set. We will refer to this expansion as a planar non-orthogonal expansion (NOE). This nonzero overlap between the basis functions means it is possible to decompose two very similar correlation functions using dramatically different coefficients, causing instability. One way to see this instability is that the inverse of $M$ (which is needed when estimating the best choice for $c_m$) has rapid sign changes and many elements that are orders of magnitude larger

---

[1]Note that the 1/4 in the exponential $e^{-r^2/4}$ means that the expansion cannot be thought of as expanding in $|\phi_m(r)|^2$.

than unity. In the example given in the figure, many coefficients take values around $10^8$. There is further discussion of this instability in Appendix D.

The core issue is that the expansion proposed by Girvin does not constitute a set of orthogonal functions (for any positive measure), since all the individual functions are positive definite.

## 3 Spherical orthogonal basis

We now construct an expansion of $g$ into orthogonal functions. To achieve this, we first move to the spherical geometry, and construct an expansion there. The planar expansion is then obtained in the limit of an infinitely large sphere.

The spherical geometry is commonly used in the literature, especially for numerical studies of the Fractional Quantum Hall Effect, and was introduced in Ref. [21]. An expansion in this geometry should therefore be of independent interest. We construct an expansion basis adapted to the sphere, in two steps. Firstly, to account for the finite geometry of the sphere, we base the expansion on spherical single-particle wavefunctions instead of planar ones. This leads to an non-orthogonal expansion analogous to Girvin's. Secondly, we orthogonalize the basis, which will remedy the above-mentioned problems with instability. At the end of this section, we take the limit of an infinitely large sphere to produce an orthogonal expansion suitable for the planar geometry.

We begin with a small primer on Fractional Quantum Hall wave functions on the sphere. The electrons are placed on a sphere of radius $R$, where the magnetic field emanates from a central Dirac monopole. Demanding that the number of magnetic flux quanta piercing the surface is an integer $2Q$ leads to the relationship

$$R = \ell \sqrt{Q}, \tag{9}$$

where $\ell = \sqrt{\hbar/eB}$ is the magnetic length. On the sphere there is a geometrical shift $S$ in the relationship between the number of electrons $N$, flux $2Q$ and filling factor $\nu$, defined by

$$2Q = N/\nu - S. \tag{10}$$

The shift is a topological quantum number (for homogenous states), and can be used to distinguish different topological states residing at the same filling fraction.

We define a dimensionless distance $\eta$ between two particles at positions $\boldsymbol{r}_1$ and $\boldsymbol{r}_2$ as

$$\eta = \frac{|\boldsymbol{r}_1 - \boldsymbol{r}_2|}{2R} = \frac{r}{2R}, \tag{11}$$

where $r$ is the chord distance (through the interior of the sphere) between the two particles. It is often convenient to use spinor coordinates $u$ and $v$ on the sphere, related to the polar and azimuthal angles $\theta$ and $\phi$ by $u = \cos(\theta/2)\exp(i\phi/2)$ and $v = \sin(\theta/2)\exp(-i\phi/2)$. In terms of these, we can write $\eta = |u_1 v_2 - u_2 v_1|$. Similarly to Girvin's planar version, we would like the expansion to consist of the $\nu = 1$ pair correlation function $g_1(\eta)$ plus a superposition of basis functions. The function $g_1$ is given as (see appendix A):

$$g_1(\eta) = 1 - (1 - \eta^2)^{2Q}. \tag{12}$$

Following a procedure similar to Girvin's [5] then gives the following spherical expansion

(see appendix B):

$$g(\eta) = 1 - \left(1 - \eta^2\right)^{2Q} + \sum_{k=1}^{2Q} d_k f_k(\eta),$$

$$f_k(\eta) = \sqrt{\frac{4Q+1}{4\pi}\binom{4Q}{2k}}(1-\eta^2)^{2Q-k}\eta^{2k}. \tag{13}$$

This is a finite basis due to the finite dimensionality of the Landau levels on the sphere. The functions $f_k$ are normalized under the measure $dS = 8\pi\eta d\eta$ and the inner product over $\eta$ is defined as

$$\langle f|g \rangle = 8\pi \int_0^1 d\eta\, \eta\, f(\eta)\, g(\eta)\,. \tag{14}$$

Just like on the plane, these functions are not orthogonal, but have the positive definite inner products

$$\langle f_m|f_n \rangle = \binom{4Q}{m+n}^{-1}\sqrt{\binom{4Q}{2m}\binom{4Q}{2n}}. \tag{15}$$

This expansion is exact, as long as the pair correlation function is isotropic and homogeneous. We are interested in extracting the expansion coefficients in the thermodynamic limit, and we note that the basis automatically scales in size like the quantum Hall system. The basis functions are localized at roughly the same values of $r$ (not $\eta$) irrespective of the system size. As more basis functions are added (for larger $Q$), one can probe details in the pair correlation function at larger separation between the particles. We mention in passing that one may, in principle, use a value of $2Q$ that does *not* correspond to the strength of the Dirac monopole, but then the above mentioned properties may be lost.

Orthogonalizing the functions $f_k$ through the Gram-Schmidt procedure with respect to the integration measure $dS = 8\pi\eta d\eta$ yields the following expansion in a basis of orthogonal polynomials (see appendix C for details)

$$g(\eta) = 1 - (1 - \eta^2)^{2Q} + \sum_{n=1}^{2Q} c_n G_n(\eta),$$

$$G_n(\eta) = \mathcal{N}_n \eta^2 (1 - \eta^2)^{2Q-n} J_{n-1}^{(2,4Q+1-2n)}(1 - 2\eta^2),$$

$$\mathcal{N}_n = \sqrt{\frac{(4Q+2-n)(4Q+1-n)(4Q-2n+1)}{4\pi Q n(n+1)}}\,, \tag{16}$$

with the Jacobi polynomials $J_k^{(\alpha,\beta)}(x)$ [22]. The explicit form used in our calculations is thus

$$G_n(\eta) = (-1)^n \mathcal{N}_n \eta^2 \sum_{s=0}^{n-1}\binom{n+1}{n-1-s}\binom{4Q-n}{s}\left(-\eta^2\right)^s\left(1-\eta^2\right)^{2Q-1-s}. \tag{17}$$

The overall factor of $(-1)^n$ is inserted for later convenience in order to give $c_n$ a smooth envelope shape. Some of the orthogonal basis functions are plotted in figure 2. The basis functions have three properties, which can be identified in the image: 1) The number of oscillations, i.e., maxima, of the function $G_n$ is precisely $n$. 2) The position of the last maxima of $G_n(\eta)$ is approximately located at the peak position of $f_n(\eta)$, and that distance grows as $\sqrt{n}$. 3) The distance to the first peak of $G_n(\eta)$ decreases as $1/\sqrt{\eta}$. From these three properties we conclude that our basis has the peculiar property that larger $n$ is modeling $g(\eta)$ at both shorter and longer distances simultaneously.

Even though (16) is an exact expression for $G_n(\eta)$, in practice it is not possible to accurately evaluate it numerically if $n$ is too large, $n \gtrsim 35$. The reason for this loss of accuracy is

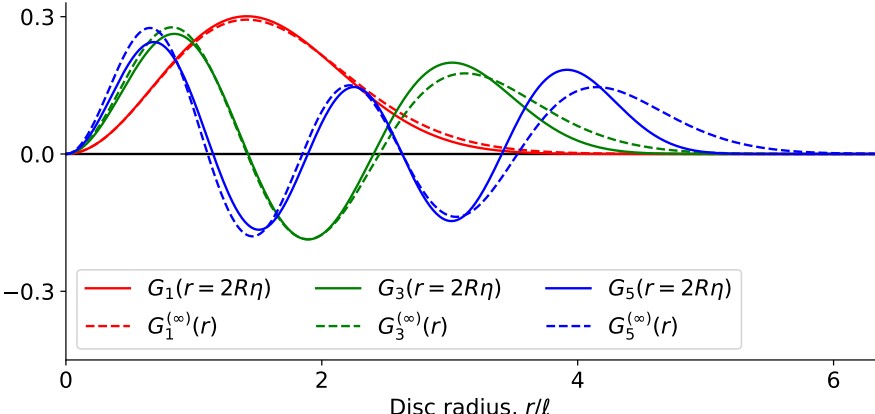

Figure 2: *Dashed lines:* The first 3 odd orthogonal basis functions in the thermodynamic limit $G_n^{(\infty)}(r)$ (19). *Solid lines:* The same basis functions $G_n(\eta)$ given by equation (16), is plotted for $2Q = 20$. The coordinate $r = 2R\eta$ with $2R^2 = 2Q = 20$ has been employed in order to compare with the infinite functions.

the almost canceling positive and negative contributions in the sum. These can, in turn, be traced back to the instability of the original expansion basis demonstrated in figure 1. This is essentially the same phenomenon as that described in Figure 1. Instead, we evaluate $G_n(\eta)$ by numerically integrating the appropriate second-order differential equation, which is much more well behaved (see Appendix E).

## 3.1 The planar limit

One goal of constructing an orthonormal set of basis functions is to enable scaling of the coefficients to the limit $N \to \infty$. This is only meaningful if the corresponding limits of the functions in (16) exist, which indeed they do. In practice, we use a subset of the functions by imposing a cutoff $K$ so that $n \in \{1, \ldots, K\}$.

From (9) and (10) the limit implies $2Q \to \infty$, so that the radius of the sphere becomes infinite and the geometry approaches a plane. Then writing $2Q \approx N/\nu$ and reverting to the chord length $r$ through $\eta = \frac{r}{2R} \approx \frac{r}{\sqrt{2N/\nu}}$, we find that the non-orthogonal basis has

$$\lim_{N \to \infty} f_k \propto e^{-r^2/2} r^{2k} \propto h_k(r). \qquad (18)$$

The orthogonalized functions inherit this limit and become

$$G_n^{(\infty)}(r) = \lim_{N \to \infty} G_n(\eta) = (-1)^n \frac{e^{-r^2/2} r^2}{\sqrt{\pi n(n+1)}} L_{n-1}^2\left(r^2\right), \qquad (19)$$

where $L_t^s(x)$ are the associated Laguerre polynomials [22]. The functions $G_n^{(\infty)}$ are orthonormal with respect to the planar integration measure (7) given by $dS = 2\pi r \, dr$.

The astute reader will notice that the expansion by Girvin on the plane only used odd powers of $r^2$, whereas here we produce also the even powers. The reason for this discrepancy is that Girvin used a center of mass frame, where both particles were an equal distance from the origin. To preserve the fermionic nature, the expansion has to only contain odd powers of $r^2$. In our expansion we first explicitly split off the antisymmetric behavior of the wave function for particles 1 and 2 and then break the symmetry between particles 1 and 2 by placing particle 1 at the origin, or the north pole of the sphere before we take the planar limit,

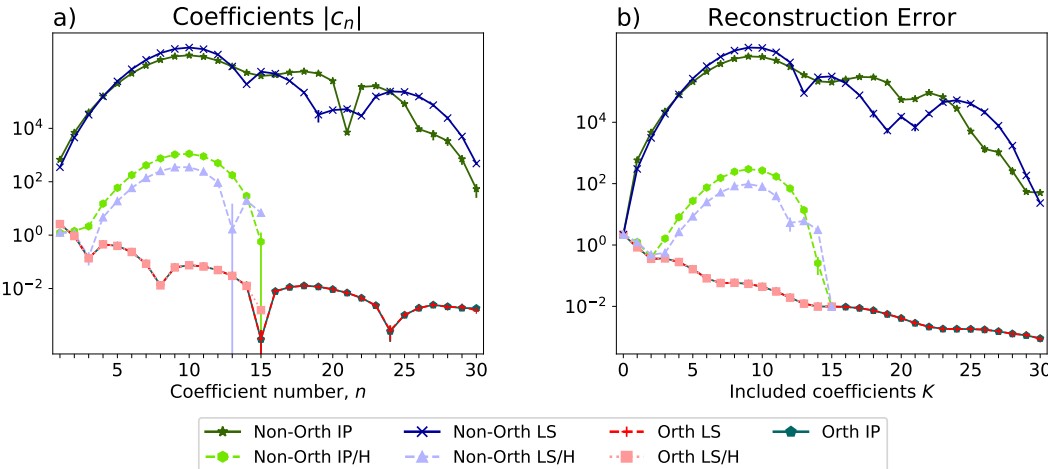

Figure 3: Expansion of the Laughlin $\nu = 1/3$ state at $N = 22$ electrons, $N = 6 \times 10^6$ samples onto $K_{\max} = 30$ orbitals, using the non-orthogonal (13) and orthogonal spherical (16) expansions. To indicate the coefficient (in)stability, we also perform the same exercise with half of the orbitals ($K_{\max} = 15$, marked with "/H" in the legend). Note the abbreviations IP (inner product) and LS (least squares) for the method used to obtain the coefficients.

a) Absolute values of expansion coefficients. The NOE coefficients $d_k$ are orders of magnitude larger than the OE coefficients and depend strongly on $K_{\max}$ and the method to find them. The OE coefficients, on the other hand, are stable, and show clear decreasing magnitude with $n$.

b) The measure $\epsilon$ as defined in (20) versus the number $K$ of functions included in expansion. For the OE, the reconstruction becomes monotonically better with more included coefficients. For $K_{\max} = 15$, the NOE reconstruction becomes worse at intermediate $K$, only to become as good as the OE at $K = K_{\max}$. For $K_{\max} = 30$, the NOE reconstruction immediately worsens and never recovers, indicating its lack of stability.

see Eq. (B.1) and (B.2) in Appendix B. This yields a different expansion with both odd and even powers of $r^2$. This may seem inefficient but we need to keep in mind that our $r$ is the actual distance between the particles whereas Girvin's $r$ is only half that distance, as it gives the distance to the joint centre of mass. Therefore the expansions need the same number of nonzero coefficients to model $g$ out to the same actual distance.

## 4 Benchmarking the expansions

We now benchmark how the new orthogonal expansion (OE) in equation (16) performs compared with the non-orthogonal expansion (NOE) in equation (13) and (4). An expansion should be robust in terms of the coefficients, not only with respect to perturbations and the number of basis functions but also to the chosen method of finding the coefficients. We will use two different methods to obtain expansion coefficients and compare them.

The first method consists of a straightforward least squares (LS) fit of the Monte Carlo approximated bin averages to the relevant basis functions. To be precise, the LS method min-

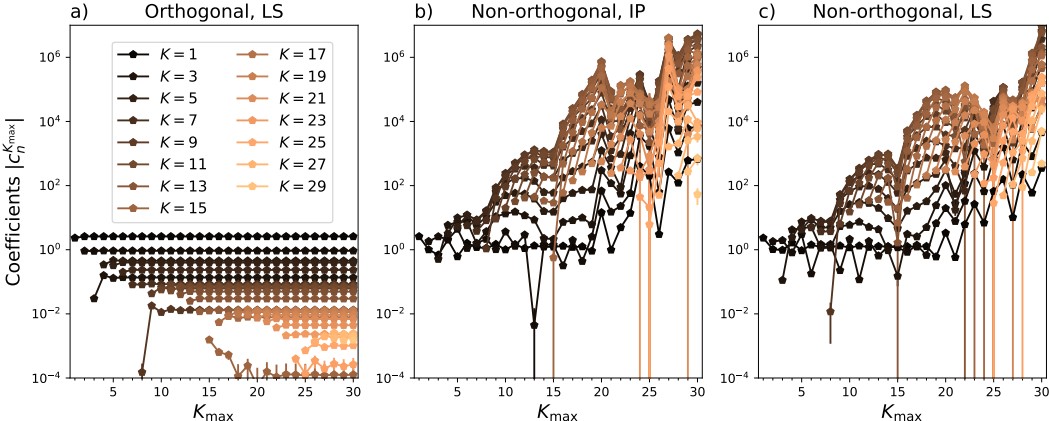

Figure 4: Evolution of the pair correlation expansion coefficients $c_n^{K_{max}}$ when a number $K_{max}$ are included. The orthogonal coefficients computed though the least squares method (a) quickly stabilize whereas the nonorthogonal coefficients depend on $K_{max}$ tend to increase as $K_{max}$ increases.

imizes the cost function

$$\epsilon = \sqrt{\sum_j \left(g(\eta_j) - g_{MC}(\eta_j)\right)^2}, \tag{20}$$

where $g$ is the NOE/OE expansions in (13) and (16), and $g_{MC}$ is the target pair correlation function approximated with bins as in (3). The $\eta_j$ are the midpoints of the bins. The least square fits are approximating $g_{MC}$ by minimizing $\epsilon$, which also is an expression for the reconstruction error.

The second way to find the coefficients uses the inner product (IP) method

$$c_n = \langle G_n | g_{MC} - g_1 \rangle, \tag{21}$$

where $g_1$ is the spherical pair correlation function (12) for $\nu = 1$. The IP method can be adjusted for use with the non-orthogonal basis to read

$$d_m = \sum_{k=1}^{K_{max}} (M^{-1})_{mk} \langle f_k | g_{MC} - g_1 \rangle, \tag{22}$$

where $M_{mk} = \langle f_m | f_k \rangle$ is the overlap matrix between the first $K_{max}$ functions in (13).

We now have four approaches to decomposing the pair correlation function: using the non-orthogonal spherical basis (NOE) and the orthogonal spherical basis (OE) with either of the methods least squares (LS) and inner products (IP). Here, both the LS and the IP method use pre-binned values for $g_{MC}$, so the main difference is that the two methods use different measures. For LS the measure is $dS \propto d\eta$ and for IP it is $dS \propto \eta \, d\eta$.

By making use of (19) in place of (17) a similar benchmarking is also possible directly on the plane, but we will not pursue that in this work.

## 4.1 Coefficient dependence on $K_{max}$

As a demonstration we decompose the Laughlin $\nu = 1/3$ state [23] at $N = 22$ electrons and $2Q = 3N_e - 3 = 63$, using $40 \times 10^6$ samples. Since the sphere is a finite geometry, there is an upper bound to $K_{max} \leq 2Q$, as seen in (16). We choose two different cutoffs, $K_{max} = 15$ and

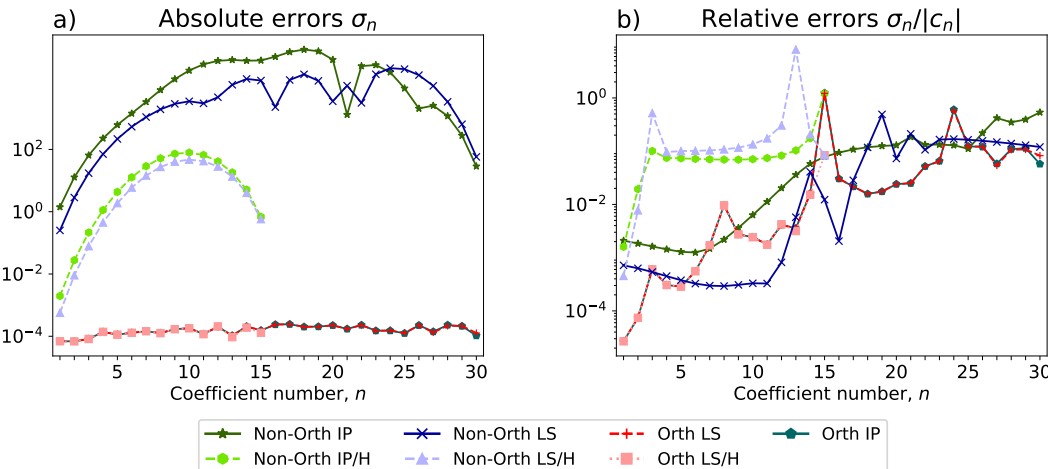

Figure 5: Expansion of the Laughlin $\nu = 1/3$ state at $N = 22$ electrons, into the same coefficients as in figure 3.
a) Absolute MC errors $\sigma_n$ of the coefficients $c_n$. The absolute errors in the OE are almost constant, whereas the NOE has errors that are several orders of magnitude larger and vary over several orders of magnitude. The larger absolute errors reflect the larger magnitude of the NOE coefficients.
b) Relative MC errors $\sigma_n/c_n$ of the coefficients $c_n$. The relative errors of the OE reflects the size of the coefficients $c_n$. As the coefficients become smaller, the relative errors become larger. For the NOE the relative errors on the coefficients are considerably larger when $K_{\max}$ is reduced.

$K_{\max} = 30$, and the corresponding expansion coefficients are plotted for the orthogonal and non-orthogonal expansion in figure 3a.

It is immediately striking that the orthogonal (OE) coefficients are independent of the method used to find them, in contrast to the non-orthogonal (NOE) ones. The latter's size also increases with $K$, rather than decreases – note the logarithmic scale. An important observation here is that if one changes the number $K_{\max}$ of included orbitals, this does not affect the OE coefficients obtained using the LS method. (The OE coefficients obtained using the IP method are also clearly unaffected by the choice of $K_{\max}$.) It does, however, change the size of the NOE coefficients by several orders of magnitude. This observation clearly indicates the non-stability of using the non-orthogonal expansion.

We have noted that changing $K_{\max}$ can profoundly impact the NOE coefficients, and now we look at this in more detail. We now study explicitly how the expansion coefficients depend on $K_{\max}$. In figure 4, we plot the evolution of the different coefficients $c_n$ as $K_{\max}$ is increased up to $K_{\max} = 30$. This is done using LS on both the orthogonal and non-orthogonal functions as well as IP for the non-orthogonal functions. The IPs of the orthogonal functions are, of course, independent of $K_{\max}$.

The non-orthogonal coefficients depend strongly on $K_{\max}$, both the least-squares and for the inner product method. In fact, the data in figure 4b-c suggests that most of the coefficients will continue to grow as $K_{\max}$ increases.

The orthogonal coefficients in figure 4a on the other hand, show much better convergence properties. As a rule of thumb, one may say that while the OE coefficients $c_n$ are converged for all $n \leq K_{\max} - 3$, the NOE coefficients are not converging at all.

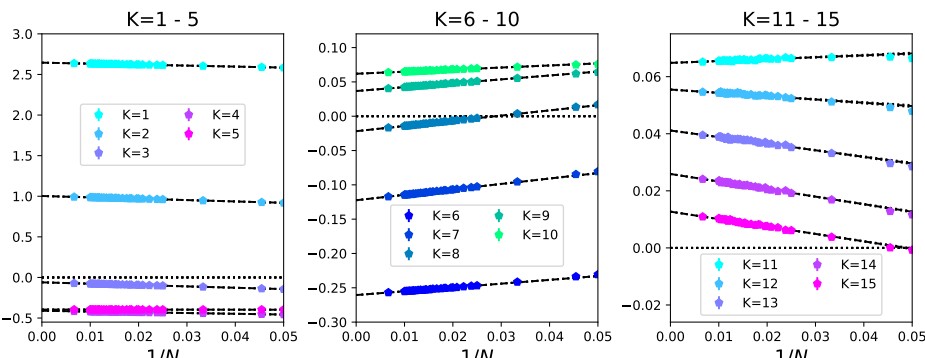

Figure 6: Expansion coefficients of the Laughlin $\nu = 1/3$ wavefunction in the orthogonal spherical expansion, obtained using inner products, plotted against $1/N$. The first 15 coefficients are shown together with linear extrapolation in $1/N$. The various coefficients have almost perfect linear scaling in $1/N$, allowing the thermodynamic limit to be determined with high accuracy.

## 4.2 Reconstruction error

Another measure we investigate is how well the various expansions $g$ manage to recreate the target correlation function $g_{MC}$. For this, we may use the residual $\epsilon$, as defined in (20). There are several potential errors that prevents us from reaching $\epsilon = 0$. The first is the statistical error due to the finite number of MC samples used to generate $g_{MC}$. The second is the binning error introduced by discretizing the correlation function into bins. On top of that, there are machine precision errors in the evaluation of the expansion functions and inverses of their overlaps. The last error is the main source of error for the NOE.

In figure 3b we plot how $\epsilon$, develops as more and more components are included in the expansions (13) and (16). This is done for fixed $K_{max}$, using the coefficients shown in figure 3a. There are several interesting things to note here. First, we see that $\epsilon$ decreases monotonically for the OE expansion, whereas adding more terms will initially make the fit worse for the NOE expansion. That the fit is not perfect, is partially due to the cutoff in $K_{max}$, but also the truncation error introduced when binning the original Monte Carlo data, as well as the bin-fluctuations introduced by the Monte Carlo procedure itself.

Further, we note that when all $K \leq K_{max} = 15$ have been included, the fit error $\epsilon$ is the same for the OE and NOE expansions. This is natural since when all $K \leq K_{max}$ are included, the two expansions should be equivalent – remember that the OE is just an orthonormalized version of the NOE. This also means (since the orthogonal IP expansion is independent of $K_{max}$) that the (blue) orthogonal IP curves are also the best approximation achievable by including the first $K$ non-orthogonal functions. The same thing is not observed in the $K_{max} = 30$ case, simply because of the numerical instability in extracting the correct coefficients when $K_{max}$ is too large.

We may understand the large reconstructing error of the non-orthogonal expansion from the arguments presented in Figure 1. More formally, we may consider the condition number of the overlap matrix $M$, which is computed in Appendix D. In the Appendix it is demonstrated that the condition number grows super-exponentially fast as a function $K_{max}$, leading to reduced numerical stability. From the above considerations, we can conclude that the NOE only yields good representations of the pair correlation for small $K_{max}$, whereas the OE is stable at all $K_{max}$ considered.

Table 1: The first 20 expansion coefficients $c_n$ for the pair correlation function of the Laughlin wavefunction at $\nu = 1/3, 1/5, 1/7$.

| $n$ | $\nu = 1/3$ | $\nu = 1/5$ | $\nu = 1/7$ | $n$ | $\nu = 1/3$ | $\nu = 1/5$ | $\nu = 1/7$ |
|---|---|---|---|---|---|---|---|
| 1 | 2.6449(6±3) | 3.5613(97±12) | 3.73192(5±5) | 11 | 0.0643(9±5) | −0.306(69±10) | −1.278(43±14) |
| 2 | 1.0027(4±3) | 2.8439(4±3) | 3.4647(95±19) | 12 | 0.0550(6±5) | −0.1317(2±9) | −1.150(80±16) |
| 3 | −0.0606(5±3) | 1.8254(8±4) | 3.0906(9±3) | 13 | 0.0408(3±7) | 0.0079(5±9) | −0.9593(9±7) |
| 4 | −0.4104(0±5) | 0.7141(2±4) | 2.4537(1±3) | 14 | 0.0257(4±6) | 0.110(52±13) | −0.740(17±13) |
| 5 | −0.3951(0±5) | −0.1679(0±5) | 1.5920(8±5) | 15 | 0.0126(4±7) | 0.1778(7±8) | −0.518(20±11) |
| 6 | −0.2601(6±6) | −0.6854(6±7) | 0.6622(1±6) | 16 | 0.0028(2±9) | 0.214(22±10) | −0.309(31±10) |
| 7 | −0.1220(6±6) | −0.8784(1±4) | −0.1632(7±6) | 17 | −0.0041(4±6) | 0.2255(5±9) | −0.123(5±2) |
| 8 | −0.0216(7±9) | −0.8504(3±7) | −0.7750(8±10) | 18 | −0.0082(5±8) | 0.2181(3±10) | 0.033(94±12) |
| 9 | 0.0365(8±7) | −0.7015(7±7) | −1.1434(5±9) | 19 | −0.0101(1±6) | 0.197(27±11) | 0.161(95±15) |
| 10 | 0.061(48±10) | −0.5051(7±9) | −1.2941(4±5) | 20 | −0.0102(8±7) | 0.1676(8±7) | 0.260(29±14) |

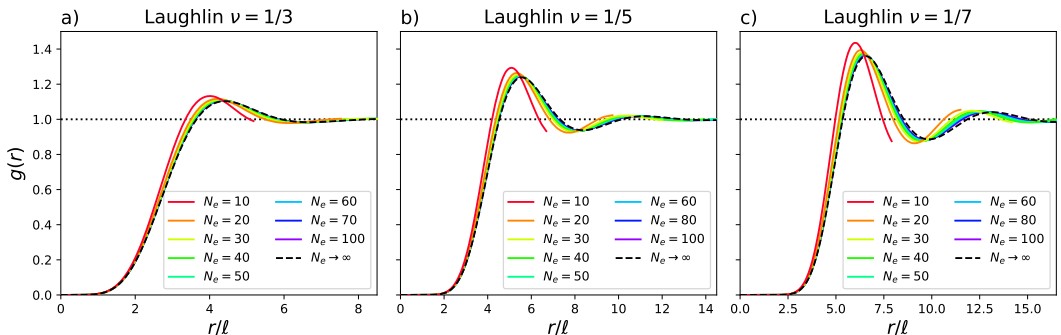

Figure 7: Pair correlation functions at finite sizes and in the thermodynamic limit for the Laughlin wavefunctions at a) $\nu = 1/3$ b) $\nu = 1/5$ and c) $\nu = 1/7$.

## 4.3 Monte Carlo Convergence

We now turn to the Monte Carlo convergence of the coefficient expansion for a fixed $K_{\max}$. The Monte Carlo errors $\sigma_n$ of individual coefficients are plotted in 5a, and the relative errors in 5b. The absolute errors in the orthogonal expansion are hovering at an almost constant $10^{-4}$, whereas the NOE expansion has absolute errors that are several orders of magnitude larger. The larger absolute errors for the OE are, however, mostly a reflection of the larger size of the coefficients (see figure 3). The relative errors of the OE reflects the size of the coefficients $c_n$. As the coefficients become smaller, the relative errors become larger. For the NOE the relative errors on the coefficients are considerably larger when $K_{\max}$ is reduced. The relative Monte Carlo errors on the NOE coefficients are not a good reflection of their (lack of) ability to reconstruct the pair correlation function.

## 4.4 Thermodynamic Limit Extrapolation

Finally (and maybe most importantly), we wish to confirm that the expansion coefficients have a well-defined thermodynamic limit. Figure 6 shows the first 15 orthogonal coefficients obtained through inner products, plotted against $1/N$ for system sizes up to $N = 150$, all using $40 \times 10^6$ Monte Carlo points. Linear (dash-dotted line) extrapolations in $1/N$ are superimposed. As the figure clearly shows, the various coefficients have an almost perfect linear scaling in $1/N$, allowing the thermodynamic limit value for the coefficients $c_n$ to be determined with high accuracy. As we have argued earlier in section 4.1, obtaining good limits using the non-orthogonal expansion is not even a converging problem.

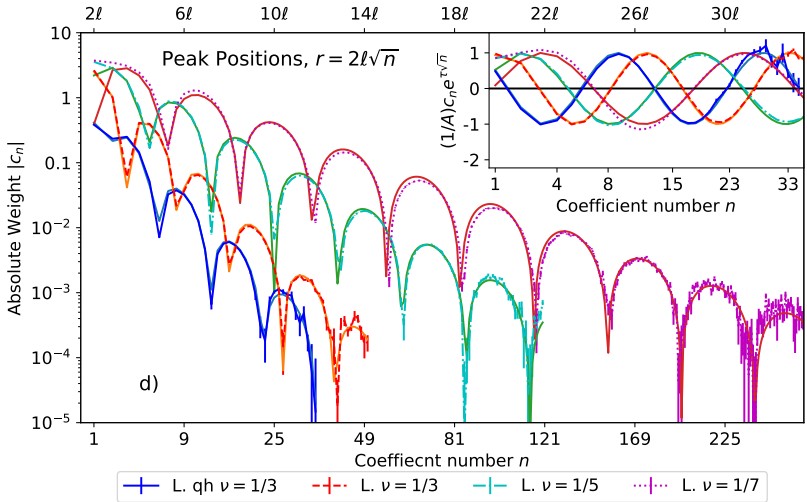

Figure 8: Absolute value of coefficients for the wavefunctions scaled to the thermodynamic limit. Note the horizontal scale proportional to $\sqrt{n}$ to emphasize the regular structure of the coefficients sizes. The top axis shows the corresponding peak positions $r = 2\ell\sqrt{n}$ for each component $r^{2n}e^{-r^2/4\ell^2}$ in equation (4). Note the superimposed best fit of equation (23), which is remarkably accurate.
*Inset*: The coefficients $c_n$ rescaled by $e^{\tau\sqrt{n}}/A$ to highlight the periodic behavior in $\sqrt{n}$. Also here the best fit is superimposed.

## 5   Pair correlations

We now apply the orthogonal spherical expansion basis, to scale the pair correlation functions, of some of the most prominent trial wavefunctions, to the thermodynamic limit: The Laughlin wavefunction [23–27] at $\nu = 1/3$, $1/5$ and $1/7$, Composite Fermions [28–31] at $\nu = 2/5$ and $\nu = 3/7$, and reverse flux CF [32–34] at $2/3$ and $3/5$. We also consider Moore-Read [35–38] at $\nu = 5/2$ and Bonderson-Slingerland [39,40] at $\nu = 12/5$.

The number of basis functions $2Q$ at a given system size $N$ goes to infinity in the macroscopic limit. However, since quantum Hall effect states have the property that $g(\eta) \to 1$ when $\eta$ is much bigger than the correlation length, the number of functions required for a good description of the pair correlation function is limited.

The first 20 coefficients for the Laughlin $\nu = 1/3$, $\nu = 1/5$ and $\nu = 1/7$ states, scaled to the thermodynamic limt, are displayed in table 1 together with their error estimates. The resulting pair correlation functions are plotted together with the finite system versions in figure 7. For these three systems we see a smooth progression of pair correlations with the number of particles $N$. The functions for individual system sizes are clearly converging towards the thermodynamic limit, with decreasing differences between lines corresponding to increasing system size. A longer table for $\nu = 1/5$ and $\nu = 1/7$ is shown in Table 7 in Appendix H.

The reach of basis function number $n$ in equation (16) is $\eta \propto \sqrt{n}$, which means that the required number of coefficients depends on how much of the length of the pair correlation function we want to model. It is natural that adding more coefficients always gives a better fit, but we note that the improvements come in steps (again see figure 3b). The midpoints of these steps correspond to the pronounced minima separating certain groups of coefficients in both Figure 3a and Figure 8. One may roughly think of the number of improvement steps as corresponding to the number of (half) oscillations in the pair correlation function compared with the $g_1$ background. Indeed, by looking at the inset in Figure 8 one can see how $c_n$ roughly

Table 2: Best fit parameters to the anzats $c_n \approx A e^{-\sqrt{n}\tau} \cos(\sqrt{n}\omega + \delta)$ in (23). For the first few coefficients the fit is not so good, but then it is excellent - especially for the Laughlin series.

| State | $A$ | $\tau$ | $\delta$ | $\omega$ |
|---|---|---|---|---|
| Laughlin $\nu = \frac{1}{3}$ | 12.91 | 1.56 | 3.20 | 2.71 |
| Laughlin $\nu = \frac{1}{5}$ | 10.23 | 0.89 | 3.05 | 2.21 |
| Laughlin $\nu = \frac{1}{7}$ | 7.93 | 0.59 | 2.88 | 1.92 |
| RCF Laughlin $\nu = \frac{1}{3}$ | 9.33 | 1.35 | 2.90 | 2.86 |
| CF $\nu = \frac{2}{5}$ | 5.54 | 1.14 | 2.75 | 3.20 |
| BS $\nu = 2 + \frac{2}{5}$ | 4.44 | 0.96 | 3.61 | 2.08 |
| MR $\nu = \frac{1}{2}$ | 4.86 | 1.34 | 3.24 | 2.52 |
| Laughlin $\nu = \frac{1}{3}$ qh | 4.01 | 1.60 | 4.67 | 2.65 |
| MR $\nu = \frac{1}{2}$ $q = \frac{1}{2}$ ab. qh | 2.48 | 1.36 | 4.79 | 2.47 |
| MR $\nu = \frac{1}{2}$ $q = \frac{1}{4}$ n-ab. qh | 1.59 | 1.41 | 4.53 | 2.54 |

follows a damped exponential of the form

$$c_n \approx A e^{-\sqrt{n}\tau} \cos(\sqrt{n}\omega + \delta), \tag{23}$$

where $\tau$ is the half-life and $\omega$ is the oscillation frequency. To bring out the oscillatory shape, $c_n$ is scaled by our best estimate of $e^{\sqrt{n}\tau}/A$. The oscillations with frequency $\omega$ then correspond directly to the steps in improvements discussed above.

We tabulate the best fit of the coefficients $A$, $\tau$, $\omega$, and $\delta$ in table 2. The best fit is further plotted on top of the coefficients in Figure 8. It should be evident from the figure that this 4-parameter fit does a remarkable job of fitting the expansion coefficients. There are noticeable deviations only for the very first coefficients. One can see in Figure 8, and table 2, that the larger $\nu$ means both stronger oscillations (smaller $\tau$), and longer wavelength oscillations (smaller $\omega$). The image suggests that the oscillations will never stop but become exponentially damped at long distances.

The most noticeable feature of the tabulated/graphed coefficients for the Laughlin series is that the sizes of the coefficients increase with decreasing filling fraction, $\nu$. This is consistent with the observation that these states have larger correlation holes and stronger oscillations due to the decreasing density. As the coefficients measure the deviation from the $\nu = 1$ state, which is described by $c_n = 0$ for all $n$, larger oscillations/deviations should mean larger coefficients. It is conceivable that if one pushes the filling fraction beyond $\nu^{-1} \gtrsim 70$ where the state becomes a Wigner crystal [41], that the $g(\eta \to 1) = 1$ limit will no longer be valid, causing the coefficient scaling to break down.

It is tempting to try and derive a simple expression for especially $\tau$ and $\omega$ in terms of the filling fraction $\nu$, which is the only parameter. However, for several reasons, it's unlikely that such an expression exists. Firstly, we know that there is a phase transition to a Wigner crystal at low filling, leading to potentially non-analytic behavior in $c_n$. Secondly, since the NOE basis models both long and short distance features at the same time, the coefficients $c_n$ are also sensitive to both length scales. As a result, they depend on a mix of long range behaviour, including topological features, and microscopic short-range details. We will comment more on this is the next section.

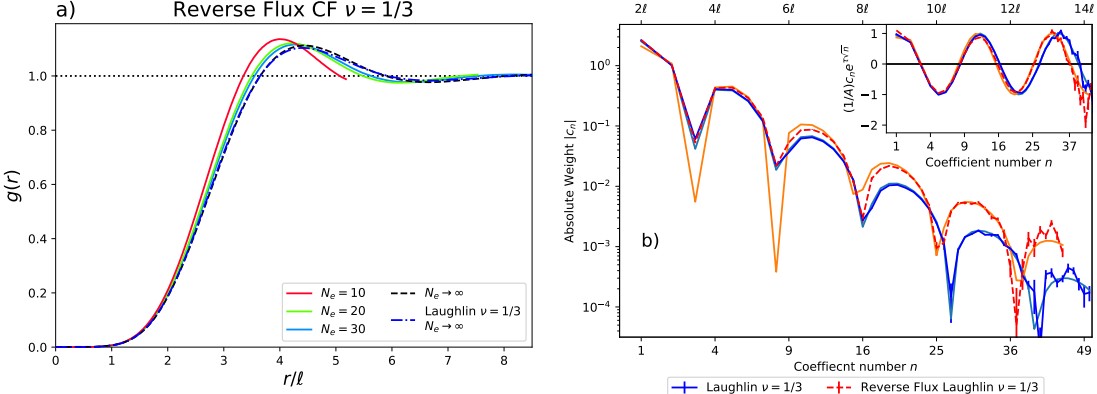

Figure 9: Pair correlation functions for the Reverse flux Composite fermion Laughlin wavefunctions at $\nu = 1/3$. This corresponds to choosing $d = 1$ in equation (24). b) The correlation coefficients for the Reverse Flux and Laughlin wavefunctions scaled to the thermodynamic limit. While the pair correlation functions a) of the two Laughlin states are almost indistinguishable in the thermodynamic limit (dashed lines vs. dotted lines), the difference can be clearly detected by comparing the expansion coefficients in b).

A technical comment is warranted: In order to resolve the coefficients $c_n$ with larger $n$ properly, we choose to scale the number of histograms bins as $N_{\text{Bins}} \propto \sqrt{2Q}$. This way the density of bins, $R/N_{\text{Bins}}$ is roughly constant as a fuction of system size $N$. Using this scaling allows us *e.g.* to accurately resolve more than 170 coeffcients for the $\nu = 1/7$ Laughlin state.

## 5.1 Systems at the same filling fracton

A direct application of our expansion is that we now can in a quantitative way compare two correlation functions at the same filling fraction.

### 5.1.1 The Laughlin family at $\nu = 1/3$

As an example of this we first consider the filling fraction $\nu = 1/3$, where there is a family of Laughlin-like wavefunctions given by

$$\Psi = e^{-\frac{1}{4} \sum_i |z_i|^2} \prod_{i<j} \left(z_i - z_j\right)^3 \left|z_i - z_j\right|^{2d} , \tag{24}$$

where $d \geq 0$ can take any real value. These functions were introduced in Ref. [42] and explored in detail in Ref. [10]. The $d = 0$ case is the normal Laughlin wavefunction, and the $d = 1$ case is a reverse flux composite fermion construction of the same state. This state is constructed by forming a $n = 1$ composite fermion lambda level whose composite fermions capture four reverse fluxes ($p = 2$). These two functions exist at the same shift and are topologically identical, but the latter one has a better overlap with the Coulomb ground state [10,43]. The pair correlation function for the (modified) Laughlin state can be seen in Figure 9a with the thermodynamic limit of the conventional Laughlin state added as a dotted line. On the scale of the plot, the difference between the two is (almost) indistinguishable. However, in Figure 9b, the difference can readily be seen in the OE coefficients.

Comparing these two wave functions allows us to gain further intuition when interpreting the coefficients $c_n$. It is known that the pair correlation function of $d = 1$ state has slightly

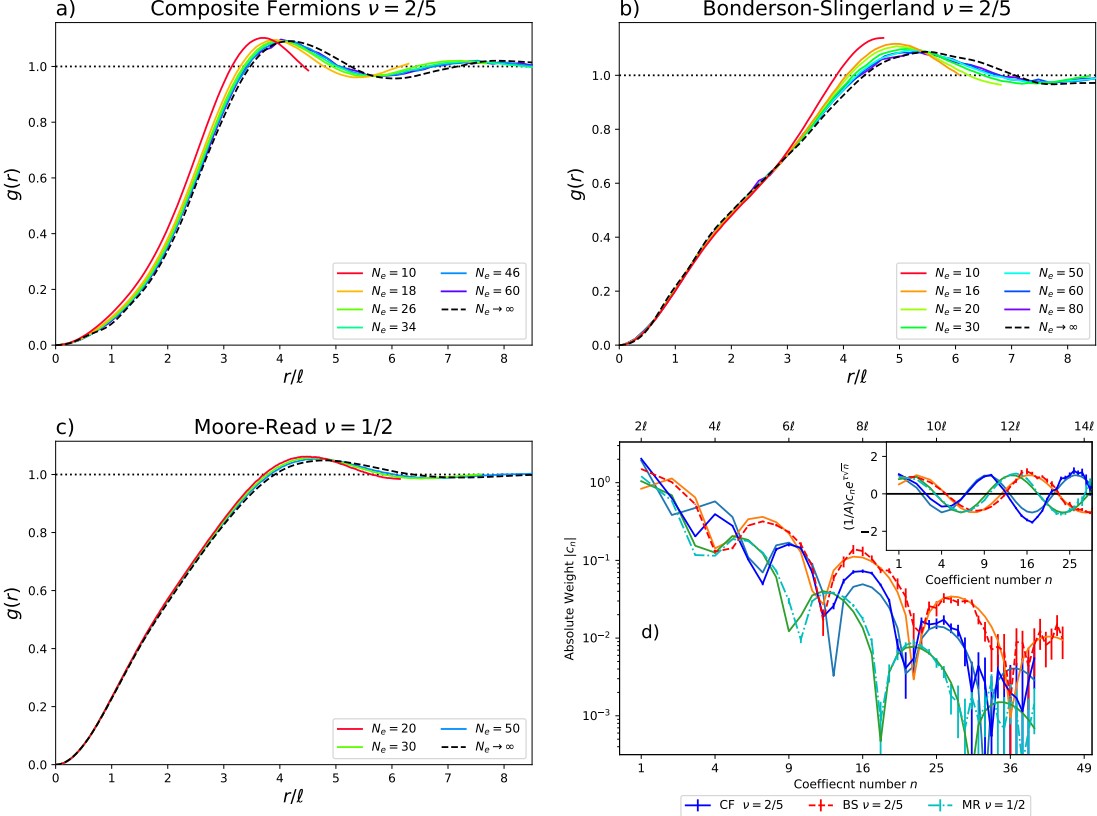

Figure 10: Pair correlation functions at finite sizes and in the thermodynamic limit for
a) the Composite fermion wavefunctions at $\nu = 2/5$, b) the Bonderson-Slingerland
wavefunctions at $\nu = 2/5$. , c) the Moore-Read wavefunctions at $\nu = 1/2$. d) The cor-
relation coefficients for the three wavefunctions scaled to the thermodynamic limit.
The graph in b) looks noticeably different from that plotted in Ref. [40] – a different
correlation function was accidentally plotted there.

stronger oscillations than the usual ($d = 0$) Laughlin state. This can be seen by looking care-
fully at Figure 9a, and is reflected also in the $d = 1$ having coefficients that decay at a lower rate
($\tau = 1.35$ as compared to $\tau = 1.56$). Both set of coefficients are well described by equation
(23), but with slightly different frequencies ($\omega = 2.86$ as compared to $\omega = 2.71$).

We note that the first handful of OE coefficients is almost the same for the two correlation
functions. At the first two dips in the graphs of the coefficients, the differences between the
coefficients for the states look large, but this is mostly due to the use of a logarithmic scale;
these coefficients are small and a small difference between them translates to a large relative
difference, which then shows up as a large difference in the logarithm. Conversely, we must
be careful not to dismiss small differences between larger coefficient, as seen on a logarithmic
scale, as they could still be relevant when reconstructing $g$. The systematic differences between
the two sets becomes visible from around $n = 9$ and becomes more pronounced at larger
$n$. We wrote earlier that the coefficients with higher indices were responsible for modeling
features at larger distances and this is also the case here. While this is true, the basis function
$G_n$ actually has support in the range between $\frac{1}{\sqrt{n}}$ and $\sqrt{n}$, so that features at both longer and
shorter distances are better modeled with inclusion of higher $n$ functions. Of course the longer
distance behaviour is often more interesting when $\frac{1}{\sqrt{n}}$ lies well inside a correlation hole. These
considerations do mean that one should be careful when attempting an interpretation of the
relative sizes of the coefficients.

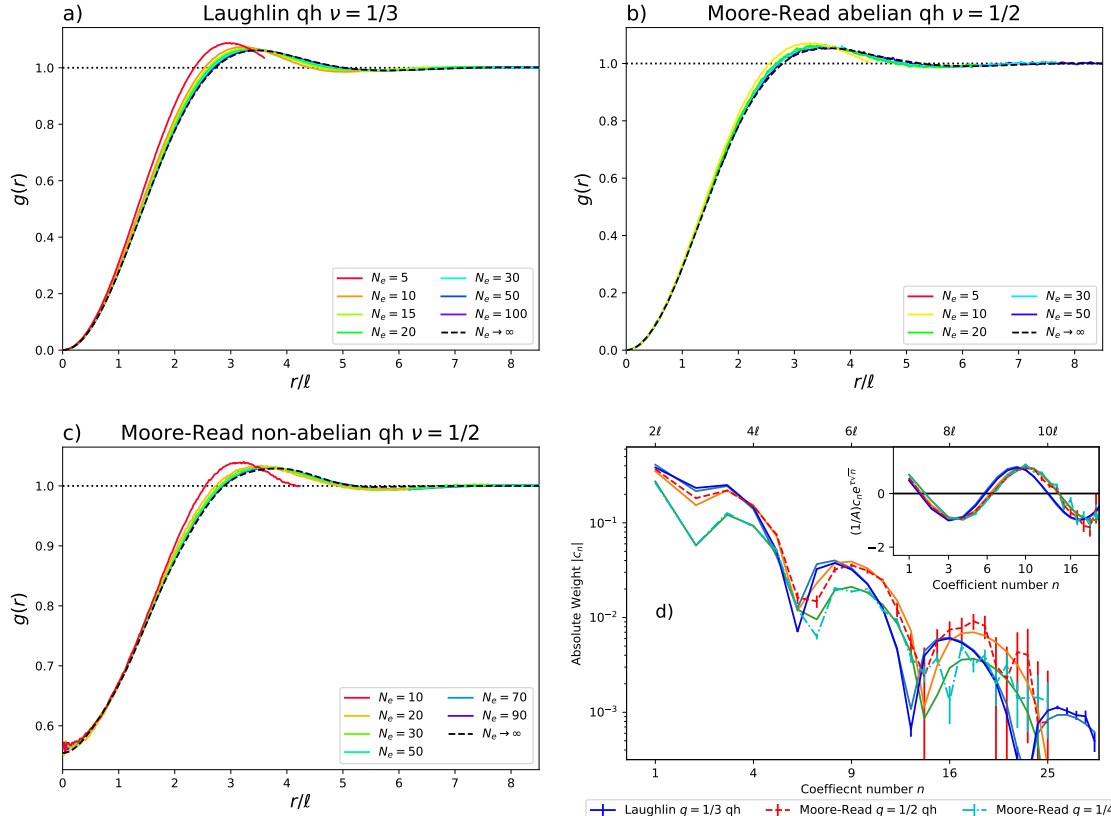

Figure 11: Density profiles of the Laughlin quasihole at $\nu = 1/3$ and abelian and non-abelian Moore-Read quasiholes at $\nu = 1/2$ for finite size and scaled systems. The to estimate the thermodynamic shape of the non abelian quasihole only correlation function data from the nother hemisphere is used, and the other non-abelian quasihole on the sourther hemisphere is ignored).

### 5.1.2 CF and BS at $\nu = 2/5$

A more interesting comparison is arguably that of the composite fermions state at $\nu = 2/5$ compared with the Bonderson-Slingerland state at $\nu = 2 + 2/5$ [39, 40]. The two states have different shifts and thus represent different topological states. However, since the shift is "sub-leading" in the number of particles in the thermodynamic limit, these two correlation functions can be directly compared. This is done in figure 10. Visual inspection allows us to see the differences between the two correlation functions easily. While the CF state has a correlation hole that is reminiscent of the Laughlin state, the BS state displays a "shoulder" at $r \sim 1.5\ell$, much in similarity with the shoulder that is found in the Moore-Read state (see Figure 10c).

That these two pair correlation functions are different can also be seen in the expansion coefficients in Figure 10d. There, one can see that the coefficients' strength and "wave-length" are different. It is perhaps not straightforward to deduce the existence of the shoulder from inspection of the coefficients. However, the first BS coefficient $c_1 = 1.53(2 \pm 5)$ is significantly smaller than the first CF coefficient $c_1 = 2.01(7 \pm 3)$ (see Table 8 in Appendix H). One can then argue that the smaller $c_1$ may remove less weight from the inner part of the correlation hole and give room for a shoulder to appear on the BS state.

## 5.2 Quasiholes in the thermodynamic limit

We now apply and adapt the above-developed formalism to expansions of quasihole densities. The density of the Laughlin quasihole [23] is closely related to the pair correlation function of the ground state. Therefore we might expect that the orthogonal expansion is also suitable for the quasihole density profile. The generated expansion coefficients confirm this for quasiholes of all states examined. We find that the thermodynamic limits are well defined and converge similarly to those in figure 6. The coefficients can be found in table 3, and a plot of the finite size and extrapolated density is shown in figure 11, for both the Laughlin $q = 1/3$ quasihole and the Moore-Read $q = 1/2$ abelian and $q = 1/4$ non-abelian quasihole. A comparison of the coefficients with the pair correlations functions for the Laughlin quasihole can also be found in fig 11d. The figure clearly shows that the Laughlin quasihole, which is smaller than the electron, also has smaller expansion coefficients.

We note that the $\tau = 1.60$ and $\omega = 2.65$ for the Laughlin quasiholes are similar to the $\tau = 1.59$ and $\omega = 2.71$ of the Laughlin state itself (see Table 2) Similarly, the abelian MR quasihole has $\tau = 1.36$ and $\omega = 2.47$, which is close to the values for $\tau = 1.34$ and $\omega = 2.52$ of the Moore-Read state itself. Thus, the values of $\tau$ and $\omega$ appear to be a property of the phase, more than the particular type of correlation function.

For the Laughlin quasiholes and the abelian Moore-Read quasiholes, the procedure for extracting the expansion coefficients is identical to that for the pair correlation functions. However, for the non-abelian Moore-Read quasiholes, the procedure is somewhat different. Here two modifications compared with the above-outlined method have been applied. The first change is that the non-abelian quasiholes come in pairs, and we choose to place the quasiholes at the north and south poles. The presence of the quasihole at the south pole will change the density at large $\eta$ and the expansion coefficients will pick up this change if one is not careful. To reduce this finite size effect, we manually set the density in the southern hemisphere to unity, i.e. we enforce $g(\eta > 1/\sqrt{2}) = 1$. This choice causes a small discontinuity in $g(r)$ on the equator, but this does not strongly affect the coefficients at lower indices.

The second difference with the abelian quasiholes (a difference that is shared with abelian quasipartices) is that the density at $\eta = 0$ is not zero. Therefore the "background" reduction $1 - (1 - \eta^2)^{2Q}$ is not a good one. We choose here to introduce an extra fit parameter $c_0$ such that the background reduction is

$$(1 - (1 - \eta^2)^{2Q})(1 - c_0) + c_0 \,,$$

with $c_0 = 1$ corresponding to the fitting used earlier. As the background is not orthogonal to the other functions in the expansion, we can not use inner products to compute $c_0$. Rather it is defined as $c_0 = g(\eta = 0)$, and estimated using a quadratic fit near $\eta = 0$. In table 3 one can see that this value in the thermodynamic limit is $c_0 = 0.55(30 \pm 15)$. There we can also see that the two abelian quasiholes have very similar early expansion coefficients (e.g. $c_1 = 0.3834(5 \pm 8)$ and $c_1 = 0.37(8 \pm 6)$), whereas the non-abelian quasihole is significantly different ($c_0 \neq 0$ and $c_1 = 0.25(95 \pm 16)$).

The similarity between the coefficients of the two abelian quasiholes can be understood in the following way. Both quasiholes enter the wavefunction (on the plane) as the same factor $\prod_i^N (z_i - \xi)$, where $\xi$ is the position of the quasihole. Thus particles are excluded from the position $\xi$ in the same way, and the difference in density at long distances enters as a sub-leading correction.

The reader should be aware that a nonzero value of $c_0$, in this case, does *not*, mean a finite probability (density) to find two overlapping fermions. It only means a finite probability for a fermion to be at the location of a non-Abelian quasiparticle. In principle, there is nothing forbidding an electron to be at the location of an Abelian quasihole, but for the wave functions

Table 3: Expansion coefficients $c_n$ for the Laughlin $\nu = 1/3$ quasihole and the Moore-Read $\nu = 1/2$ quasihole.

| $n$ | qh $\nu = 1/3$ | Abelian qh | Non-Abelian qh | $n$ | qh $\nu = 1/3$ |
|---|---|---|---|---|---|
| 0 | 0.0 | 0.0 | 0.55(30 ± 15) | 10 | 0.022(46±18) |
| 1 | 0.3834(5 ± 8) | 0.37(8 ± 6) | 0.25(95 ± 16) | 11 | 0.012(55±13) |
| 2 | −0.233(35±12) | −0.18(9 ± 4) | −0.061(2 ± 10) | 12 | 0.004(67±14) |
| 3 | −0.250(14±11) | −0.23(5 ± 6) | −0.125(5 ± 9) | 13 | −0.000(51±15) |
| 4 | −0.144(99±11) | −0.15(6 ± 6) | −0.088(0 ± 9) | 14 | −0.003(76±14) |
| 5 | −0.050(65±16) | −0.07(5 ± 4) | −0.05(15 ± 13) | 15 | −0.005(48±18) |
| 6 | 0.006(98±10) | −0.00(5 ± 6) | −0.01(11 ± 14) | 16 | −0.005(71±13) |
| 7 | 0.032(19±16) | 0.02(6 ± 6) | 0.00(69 ± 11) | 17 | −0.005(3 ± 2) |
| 8 | 0.0371(5 ± 8) | 0.04(0 ± 6) | 0.01(78 ± 11) | 18 | −0.004(4 ± 2) |
| 9 | 0.0317(4±10) | 0.03(8 ± 5) | 0.01(69 ± 13) | 19 | −0.003(06±16) |

considered here, this does not happen. The reason that this doesn't happen for the Laughlin states or the Abelian $\nu = 5/2$ quasihole (studied in this work) has to do with the fact that these holes carry a full flux quantum and require some winding in the electron wave function's phases, which then requires at least one zero in the wave function near the core of the quasihole. To see this, imagine you make a loop around the quasihole, far away from its core: To get a winding, you should consider the value of the wave function as a function of one single electron coordinate on the loop with all other electrons fixed. The phase of the wavefunction should have a winding of $2\pi$. If you shrink the loop to a point, this winding has to go down to zero, and jumps of the winding number can only happen when the loop crosses a zero of the wave function.

With paired/clustered states, it is only required that the "wave function of the pair/cluster" picks up an integer winding. Thus, in the case of the Pfaffian, if you try to bring a pair of electrons to the site of the non-Abelian quasihole, the wave function then vanishes with an extra zero compared to what you might expect already from bringing the two fermions to the same position. However, if you bring only one electron to the quasihole site, the same does not apply. Something similar also happens for the (Abelian) minimal quasiholes of the multi-Lambda level Jain states like the charge 1/5 quasihole of the $\nu = 2/5$ state. Those quasiholes would also have a nonzero probability density to find an electron at the site of the hole.

Finally, when considering the quasiholes, keep in mind that one gets considerably less precision for the same numbers of Monte Carlo samples than for the two-particle correlation functions. This is since for $N$ particles one can extract $N \cdot (N-1)/2$ relative distances, whereas one can only extract $N$ absolute positions, which is what is used for the quasihole correlation functions. Thus to obtain the same precision for the quasiholes, one needs about $N/2$ times more data points.

## 6 Estimation of Magneto-Roton Gap

Having established a stable scheme to extract correlation functions in the thermodynamic limit, we can now use this to estimate the magneto-roton gap [4, 44]using the single-mode approximation. The magneto-roton branch is often the lowest-lying neutral excitation and therefore plays a vital role in the stability of the fractional quantum Hall liquid. It gained much attention in the early literature [45, 46], and has more recently seen a revival due to the connection with gravity modes and holography [47–51]. The magneto-roton gaps have

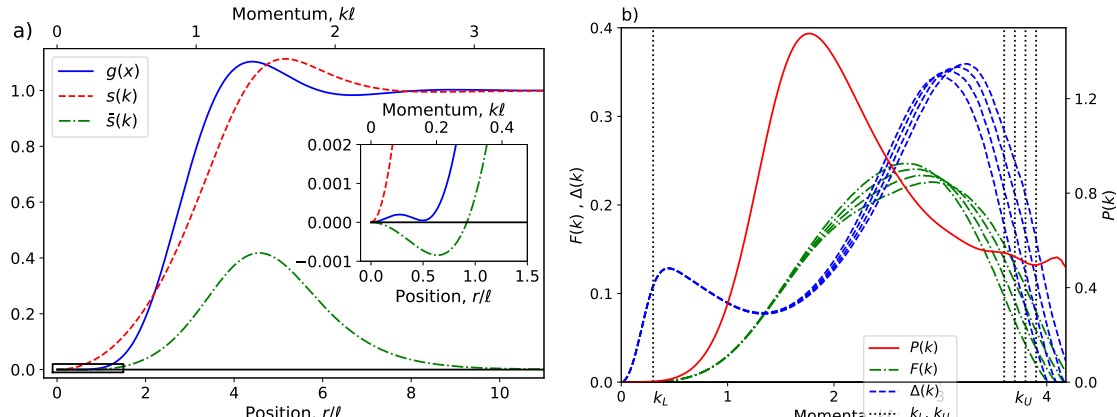

Figure 12: a) Pair correlation function $g(x)$, structure factor $s(k)$ and projected structure factor $\bar{s}(k)$ for the Laughlin $\nu = 1/3$ state in the thermodynamic limit using $K_{\max} = 30$ terms.
b) $P(k)$, $F(k)$ and $\Delta(k)$ computed using $K_{\max} = 39$ terms. Vertical lines mark the lower convergence range $k_L$ and the upper convergence range $k_U$ for the polynomial part $P(k)$ of the projected structure factor $\bar{s}(k)$. The different lines for $F(k)$ and $\Delta(k)$ illustrate the uncertainty introduced by applying the cutoff $k_U$ when computing $F(k)$ and the lines correspond to choosing $k_U + \delta k$ where $\delta k = -0.2, -0.1, 0.0, 0.1$.

been studied in detail using the composite fermion theory [52,53], finite thickness calculations [54], and recently also using DMRG [55]. Further there are recent studies of the magneto-rotons in bilayer graphene [56], nematic FQHE states [57], as well as experiments [58] and experimental proposals [59].

It is well known that the single-mode approximation will not accurately reproduce the correct shape of the magneto-roton gap, and is only expected to be correct in the long wavelength limit [60]. Nevertheless it is a good test-bed for our correlation function expansion. The original derivation of the single-mode approximation can be found in Ref. [4] and here we only recall the most important ingredients. Following Ref. [4] the magneto-roton gap is given by $\Delta(k) = \frac{\bar{f}(k)}{\bar{s}(k)}$, where $\bar{s}(k)$ is the projected static structure factor

$$\bar{s}(k) = s(k) - \left(1 - e^{-\frac{k^2}{2}}\right),$$

with $s(k)$ being the bare structure factor

$$s(\boldsymbol{k}) = 1 + \rho \int d^2\boldsymbol{R}\, e^{-\imath \boldsymbol{k} \cdot \boldsymbol{R}} g(\boldsymbol{R}), \tag{25}$$

where $\rho$ is the density of the system. The function $\bar{f}(k)$ is defined as

$$\bar{f}(\mathbf{k}) = 2 \int \frac{d^2\mathbf{q}}{(2\pi)^2} v(\mathbf{q}) \sin^2\left(\frac{1}{2}|\mathbf{q} \times \mathbf{k}|\right)\left[e^{\mathbf{k}\cdot\mathbf{q}}\bar{s}(\mathbf{k}+\mathbf{q}) - e^{-\frac{1}{2}k^2}\bar{s}(\mathbf{q})\right], \tag{26}$$

where $v(\mathbf{q}) = \frac{2\pi}{|\mathbf{q}|}$ is the Fourier transforms of the Coulomb interaction. Since $g \to 1$ at large distances $\bar{s}(k)$ has a delta function component at $\mathbf{k} = 0$. Since this delta function does not contribute to $\bar{s}(k)$ we drop it and work with $g - 1$ instead of $g$.

Assuming a spherically symmetric correlation function, the angular degree of freedom may be integrated out giving

$$s(k) - 1 = 2\pi\rho \int_0^\infty dr\, J_0(kr)[g(r) - 1] \cdot r, \tag{27}$$

Table 4: The expansion coefficients for the first terms in the projected structure factor as given in (30) in the main text. An overall factor of $1/\sqrt{\pi n(n+1)}$ has been omitted.

| $n$ | $\sum_{t=0}^{n} \lambda_t^{(n)} k^{2t}$ |
|---|---|
| 1 | $-k^2 + 2$ |
| 2 | $k^4 - 5k^2 + 2$ |
| 3 | $-\frac{k^6}{2} + 5k^4 - 10k^2 + 4$ |
| 4 | $\frac{k^8}{6} - \frac{17k^6}{6} + 13k^4 - 18k^2 + 4$ |
| 5 | $-\frac{k^{10}}{24} + \frac{13k^8}{12} - \frac{53k^6}{6} + 27k^4 - 27k^2 + 6$ |
| 6 | $\frac{k^{12}}{120} - \frac{37k^{10}}{120} + \frac{47k^8}{12} - \frac{127k^6}{6} + 48k^4 - 39k^2 + 6$ |
| 7 | $-\frac{k^{14}}{720} + \frac{5k^{12}}{72} - \frac{19k^{10}}{15} + \frac{32k^8}{3} - 43k^6 + 78k^4 - 52k^2 + 8$ |

where $J_0(x)$ is a Bessel function of the first kind and we now have $g-1$ instead of $g$.

In the thermodynamic limit the expansion for $g(r)$ reduces to

$$g(r) = 1 - e^{-\frac{1}{2}r^2} + \sum_{n=1}^{K_{\max}} c_n G_n(r), \tag{28}$$

where $G_n(r)$ was given in equation (19). Integrating the $e^{-\frac{1}{2}r^2}$ term and using that $2\pi\rho\ell^2 = \nu$, the projected structure factor after inserting (28) may be expressed as

$$\bar{s}(k) = (1-\nu) e^{-\frac{1}{2}k^2} + \nu \int_0^\infty dr\, r J_0(kr) \sum_{n=1}^{K_{\max}} c_n G_n(r) = (1-\nu) e^{-\frac{1}{2}k^2} + \nu \sum_{n=1}^{K_{\max}} c_n I_n(k),$$

where $I_n(k)$ are the Fourier transforms of $G_n(r)$. Using the structure of the Bessel function $J_0$ and $G_n$ we find that

$$I_n(k) = \int_0^\infty dr\, r J_0(kr) G_n(r) = (-1)^n \int_0^\infty dr \frac{r^3 e^{-\frac{1}{2}r^2}}{\sqrt{\pi n(n+1)}} J_0(kr) L_{n-1}^{(2)}(r^2) = -e^{-\frac{1}{2}k^2} \sum_{t=0}^{n} \lambda_t^{(n)} k^{2t},$$

has a polynomial part with the coefficients $\lambda_t^{(n)}$. The coefficients $\lambda_t^{(n)}$ are computed analytically in Appendix F, yielding the formula

$$\lambda_t^{(n)} = \frac{(-1)^n}{t!\sqrt{\pi n(n+1)}} \sum_{s=t}^{n} (-2)^{s-t} \binom{n+1}{n-s} \binom{s}{t} s. \tag{29}$$

The first few polynomials are given in Table 4.

For convenience, we define $P(k) = e^{\frac{1}{2}k^2} \bar{s}(k)$ such that the analytic projected structure factor $\bar{s}(k)$ can now be expanded in the compact form

$$P(k) = \bar{s}(k) e^{\frac{1}{2}k^2} = 1 - \nu \sum_{n=0}^{K_{\max}} c_n \sum_{t=0}^{n} \lambda_t^{(n)} k^{2t}, \tag{30}$$

where we defined $\lambda_0^{(0)} = c_0 = 1$. The definition of $c_0 = 1$ coincides with the $c_0$ introduced in Section 5.2, for the fermionic pair correlation function, and abelian quasiholes.

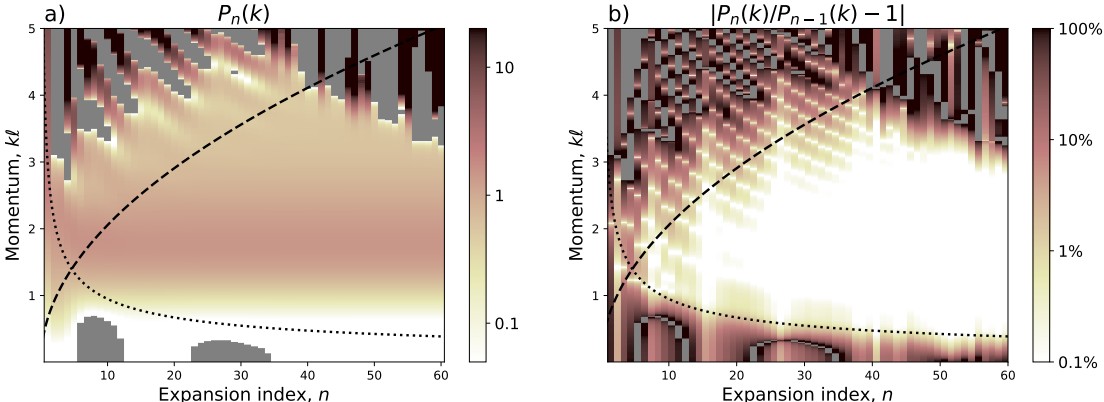

Figure 13: $n$-dependence for $P(k)$ shown as a) $P_n(k)$ and b) $|P_n/P_{n-1}-1|$ for $n \leq 60$. Dotted and dashed lines trace out how the convergence region is expected to grow as a function of $n$, if all the expansion coefficients were exact. Note that for too large $n$, convergence goes down, as the expansion coefficients $c_n$ are not resolved to enough accuracy. In a) $P_n \leq 0$ is marked with gray, and in b) $P_n/P_{n-1} \leq 0$ is marked with gray.

Table 5: Cutoff parameters used in the fitting for the Magneto-roton gap.

| Name | $k_L$ | $k_U$ | $K_{\max}$ | Name | $k_L$ | $k_U$ | $K_{\max}$ |
|---|---|---|---|---|---|---|---|
| Laughlin $\nu = 1/3$ | 0.30 | 3.8 | 39 | CF $\nu = 2/5$ | 0.50 | 3.7 | 30 |
| Laughlin $\nu = 1/5$ | 0.55 | 3.4 | 42 | CF $\nu = 3/7$ | 0.50 | 3.0 | 30 |
| Laughlin $\nu = 1/7$ | 0.45 | 3.1 | 57 | CF $\nu = 2/3$ | 0.50 | 3.1 | 33 |
| Modified Laughlin $\nu = 1/3$ | 0.4 | 3.70 | 36 | CF $\nu = 3/5$ | 0.40 | 3.5 | 39 |
| Moore-Read $\nu = 2 + 1/2$ | 0.30 | 3.4 | 31 | BS $\nu = 2 + 2/5$ | 0.60 | 3.1 | 25 |

For a correlation function that has the limit $g(r \to \infty) = 1$ then $s(0) = \bar{s}(0) = 0$. Considering equation (30), this condition takes the form of the sum rule

$$1 = \nu \sum_{n=0}^{\infty} c_n \lambda_0^{(n)} = \nu + \nu \frac{1}{\sqrt{\pi}} \sum_{k=1}^{\infty} \left( c_{2k-1} \sqrt{\frac{2k}{2k-1}} + c_{2k} \sqrt{\frac{2k}{2k+1}} \right). \quad (31)$$

In the last equality we used that $\lambda_0^{(2k)} = \frac{1}{\sqrt{\pi}} \sqrt{\frac{2k}{2k+1}}$ and $\lambda_0^{(2k-1)} = \frac{1}{\sqrt{\pi}} \sqrt{\frac{2k}{2k-1}}$. With a finite $K_{\max}$ and/or with uncertainties on the coefficients $c_n$, this equation will never be perfectly satisfied. In order to take this lack of precision into account, we may use equation (31) to solve for $\nu$. Defining $\Gamma = 1 + \sum_{n=1}^{K_{\max}} c_n \lambda_0^{(n)}$, gives the effective filling fraction $\nu^\star = 1/\Gamma$, which we can reinsert into equation (30).

The structure factor and projected structure factor can be seen extrapolated to the thermodynamic limit in figure 12a, for $K_{\max} = 30$ terms. For small values of $r$ one can see that the truncation of the approximation to $g(r)$ is not monotonic, likewise, we find that the projected structure factor dips below zero for small $k$. To get a sense of how well converged $P(k)$ is we may study $P(k)$ as a function of $K_{\max}$. In figure 13a we can see $P_n(k)$ where $n = K_{\max}$. Note the logarithmic scale, which allows to resolve both the small $k$ and large $k$ regions. Gray regions indicate where $P_n(k)$ is negative.

For $n \lesssim 35$ we can see a growing region $k_L \lesssim k \lesssim k_U$ with converged $P_n(k)$. We here observe the approximation of $P(k)$ becoming increasingly better as we add more terms for the expansion. Looking a bit closer we note that the bounds of this region scale as $k_L \propto 1/\sqrt{n}$

and $k_U \propto \sqrt{n}$. As a guide to the eye, dashed and dotted lines show the upper, $k_U$, and lower, $k_L$, approximate boundaries for the convergence. For $K_{\max} = 30$ we estimate $k_U \approx 3.5$ and $k_L \approx 0.55$. Black dotted and dashed lines trace out how the convergence region is expected to grow as a function of $n$. Beyond $n \approx 35$, the region of convergence starts to shrink, reflecting that the expansion coefficients $c_n$ are not resolved to enough accuracy.

In figure 13b we plot the absolute value of $(P_n - P_{n-1})/P_{n-1}$, where still $n = K_{\max}$. Again we use a logarithmic scale to better resolve the convergence of the low $k$ and large $k$ regions. Here, the gray regions indicate where $P_n/P_{n-1} \le 0$ and thus signal a sign change of $P_n(k)$ as a function of $n$. This figure tells the mostly same story as figure 13a, but highlights that the sweet spot where $k_L$ is minimal and $k_U$ is maximal does not seem to occur for the same $K_{\max}$. The upper limit $k_U$, has its maximal value are around $K_{\max} \approx 35-40$, whereas the lower limit is still shrinking at $K_{\max} = 60$.

The square root behavior in $k_L$ and $k_U$ for early $n$ can be qualitatively understood as follows: Inspecting $G_n(r)$ in equation (C.1) we note that the function has its last peak at a distance that scales with $\sqrt{n}$. This means that we can only model long-wavelength features to a cutoff $r_0$ that scales with $\sqrt{n}$. For the static structure factor $s(k)$ this means that the long wavelength (short $k$) bound on $k$ will scale as $1/r_0 \propto 1/\sqrt{n}$. At the same time $G_n(r)$ will undergo $n$ oscillations for $r < r_0$, meaning that $G_n(r)$ has a wavelength that scales as $r_0/n \propto 1/\sqrt{n}$. Thus we see that the shortest converged wavelength features of $g(r)$ will also converge as $1/\sqrt{n}$. This in turn means that the large $k$ convergence of $s(k)$ will scale as $k \propto \sqrt{n}$, just as observed in figure 13a. We note that the expansion presented in this work will systematically improve both the low-$k$ and high-$k$ features of $P(k)$ with increasing $K_{\max}$.

We now compute $\bar{f}(k)$ through (26). Using that we may write $\bar{s}(\mathbf{k}) = P(\mathbf{k})e^{-\frac{1}{2}k^2}$ as in (30) we find that $\bar{f}(\mathbf{k}) = e^{-\frac{k^2}{2}}F(\mathbf{k})$ where

$$F(\mathbf{k}) = 2 \int \frac{d^2\mathbf{q}}{(2\pi)^2} v(\mathbf{q}) \sin^2\left(\frac{1}{2}|\mathbf{q} \times \mathbf{k}|\right) e^{-\frac{1}{2}q^2} \left[P(\mathbf{k}+\mathbf{q}) - P(\mathbf{q})\right]. \tag{32}$$

This allows us to evaluate the magneto-roton gap

$$\bar{\Delta}(k) = \frac{\bar{f}(k)}{\bar{s}(k)} = \frac{F(k)}{P(k)},$$

in terms of the functions $P(k)$ and $F(k)$ only. Further, using that $v(q) = \frac{2\pi}{|q|}$ and changing to polar coordinates gives the function

$$F(k) = \frac{2}{\pi} \int_0^\pi d\phi \int_0^\infty dq \sin^2\left(\frac{qk}{2}\sin\phi\right) e^{-\frac{1}{2}q^2} \left[P\left(|k + e^{\iota\phi}q|\right) - P(q)\right], \tag{33}$$

which we will integrate numerically.

When integrating (33) it will be important to take into account that we really only trust $P(k)$ in the range $k_L \lesssim k \lesssim k_U$. Out of these two uncertainties, the upper limit is far more important as $P(k)$ has growing (and sign changing) oscillations, where as for small $k$ then $P(k)$ is already so small that it hardly affects the integral. However since $P(k)$ appears in the denominator of $\Delta(k)$ is has a critical impact on the result. In order to regularize $F(k)$ we impose a cutoff in the integrand such that we only use the parts of the integral where $q < k_U$ and $|\vec{k} + \vec{q}| < k_U$. Roughly speaking this implies that we should not expect $F(k)$ to be trusted for $k \gtrsim k_U/2$.

To estimate the range of validity of $F(k)$ we vary the cutoff $k_U$ around $k_U \approx 3.5$ and observe the effect on $F(k)$. The result can be found in figure 12b, with four different values of $k_U$ indicated by dotted lines. We find that the lines converge in the region $k \lesssim 1.5-2 \approx k_U/2$,

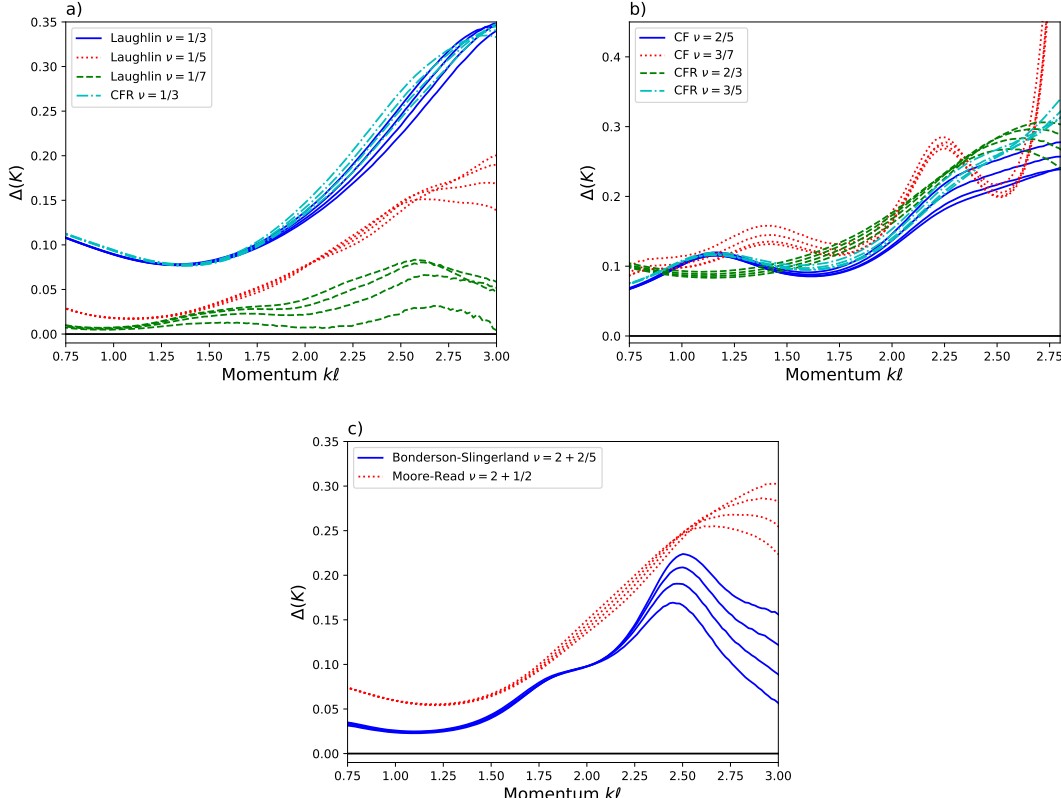

Figure 14: Estimate of the magneto-roton gap, $\Delta(k)$, from the parametrization of $g(r)$ in the thermodynamic limit.
a) Laughlin $\nu = 1/3, 1/5, 1/7$, b) (Reverse flux) Composite fermions $\nu = 2/5, 3/7, 2/3, 3/5$, c) Moore-Read and Bonderson-Slingerland at $\nu = 2 + 1/2$ and $\nu = 2 + 2/5$. The plots extend from the lower convergence range $k_L$ to the upper convergence range $k_U$ for the polynomial part $P(k)$ of the projected structure factor $\bar{s}(k)$. The different lines illustrate the uncertainty in $k_U$ when computing $F(k)$ and the lines correspond to choosing $k_U + \delta k$ where $\delta k = -0.2, -0.1, 0.0, 0.1$. The $K_{\max}$ and $k_U$ cutoff's are tabulated in table 5.

and thus that $F(k)$ can be trusted there. More details about how the truncation of the integral affect the result can be found in Appendix G, where we especially direct the reader to figure 18.

When computing $\Delta(k)$ we see that the uncertainty in $P(k)$ for small $k$ will cause $\Delta(k)$ to diverge, so too small $k$ can also not be trusted. We therefore conclude that our estimate for $\Delta(k)$ is valid in the range $.55 \lesssim k \lesssim 2$, but not much beyond that.

The good news is that one may systematically expand the range of validity of the estimate of $\Delta(k)$ by improving the estimate of the coefficients $c_k$. From figure 13 we saw that adding extra terms, even if the coefficients are small, will systematically extend the validity of $P(k)$.

We can now apply the same procedure to obtain the roton-gap for all the FQH-states considered in this work and compare it to earlier literature. This is illustrated in figure 14. The $K_{\max}$ and $k_U$ cutoff's are tabulated in table 5. For the Laughlin states in figure 14a we obtain gaps that are comparable with those obtained in Ref [4] (Fig. 4). For the Jain series, the gap is also comparable to that obtained in Ref [61] (fig 1). For the Moore-Read it is a bit smaller than reported in Ref [52], presumably due to a lack of precision in the coefficients $c_n$. Here the single-mode approximation is in any case dubious as there is also a neutral fermion mode [60, 62].

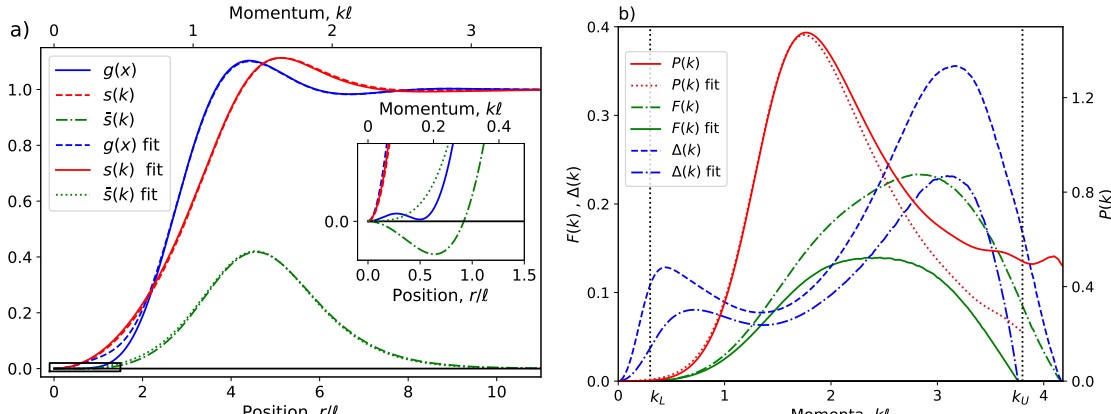

Figure 15: Comparison between the Laughlin $\nu = 1/3$ coefficients $c_n$ and the fit in equation (23). For the coefficients, the plot is the same as in figure 12. a) Pair correlation function $g(x)$, structure factor $s(k)$ and projected structure factor $\bar{s}(k)$ with $K_{\max} = 30$ terms. For $g(r)$, the difference is largest at $r \approx \ell$, giving a softer approach to zero when $r \to 0$. Also for $s(k)$ there is a significant difference when $k \to 0$. For the fit, $s(k)$ is positive for all $k$ whereas for the coefficients, $s(k)$ goes negative at around $k = 0.3$ (see figure 12 for a zoom in). Apart from that, all the main features are essentially the same. b) $P(k)$, $F(k)$ and $\Delta(k)$ computed using $K_{\max} = 39$ terms. Here, larger differences can be seen. In short, at large $k$, $P(k)$ is smaller which gives an even smaller $F(k)$. This results in a reduced $\Delta(k)$ for the fit compared to the coefficients. The difference is, however, not that dramatic, given that the fit only uses four parameters.

All in all, we find that our expansion works well. We note that the lack of precision for small $k$ mainly stemmed from an uncertainty in the structure factor $P(k)$. This can be improved upon by taking more terms into account when computing $P(k)$ for small $k$ compared to large $k$, but we have not attempted such a scheme in this work.

## 7 Using the fit to the coefficients

As a first application of the fit developed in Section 5, we will now reconstruct the pair-correlation function and compute the magneto-roton gap of the Laughlin wave function at filling nu=1/3. We will use the fit coefficients as given in table 2. The hope is that, especially at high coefficient numbers, hence at long distances, the fit coefficients will provide higher accuracy than the coefficients obtained by direct numerical calculation. At the very least, they will provide smoother behavior. In addition, if desired, the fit can be extrapolated to arbitrarily large numbers of coefficients. The numerical values of both sets of coefficients are given in table 6.

We begin with the correlation function $g(r)$, and we use the subscript "fit" for the functions constructed using the fitted coefficients. A comparison of the two versions are plotted in figure 15a) for $K_{\max} = 30$ (in blue). In the figure, for $r > 3\ell$, there is very little difference between the two functions. The largest difference occurs at around $r \approx \ell$, giving $g_{\text{fit}}(r)$ a softer exclusion region than that of $g(r)$. The maximum difference is about of 0.05 and occurs at $r \approx \ell$. This difference is consistent with the 0.12 difference between $c_1$ and $c_1^{\text{fit}}$, and the height $G_1(r \approx \ell) \approx 0.3$ of the first term in the expansion.

Table 6: The first 30 expansion coefficients $c_n$ for the pair correlation function of the Laughlin wavefunction at $\nu = 1/3$ and the fit in equation (23). These numbers are graphically depicted in figure 8.

| $n$ | $\nu = 1/3$ | Fit | $n$ | $\nu = 1/3$ | Fit | $n$ | $\nu = 1/3$ | Fit |
|---|---|---|---|---|---|---|---|---|
| 1 | 2.6449(6 ± 3) | 2.52752 | 11 | 0.0643(9 ± 5) | 0.06773 | 21 | −0.0092(0 ± 6) | −0.01001 |
| 2 | 1.0027(4 ± 3) | 1.02980 | 12 | 0.0550(6 ± 5) | 0.05761 | 22 | −0.0078(7 ± 9) | −0.00829 |
| 3 | −0.0606(5 ± 3) | −0.04169 | 13 | 0.0408(3 ± 7) | 0.04227 | 23 | −0.0059(7 ± 9) | −0.00631 |
| 4 | −0.4104(0 ± 5) | −0.39708 | 14 | 0.0257(4 ± 6) | 0.02642 | 24 | −0.0041(9 ± 5) | −0.00435 |
| 5 | −0.3951(0 ± 5) | −0.38774 | 15 | 0.0126(4 ± 7) | 0.01267 | 25 | −0.002(47 ± 11) | −0.00258 |
| 6 | −0.2601(6 ± 6) | −0.25644 | 16 | 0.0028(2 ± 9) | 0.00212 | 26 | −0.0012(8 ± 8) | −0.00109 |
| 7 | −0.1220(6 ± 6) | −0.11953 | 17 | −0.0041(4 ± 6) | −0.00507 | 27 | −0.000(37 ± 14) | 0.00006 |
| 8 | −0.0216(7 ± 9) | −0.01868 | 18 | −0.0082(5 ± 8) | −0.00926 | 28 | 0.000(81 ± 11) | 0.00090 |
| 9 | 0.0365(8 ± 7) | 0.04023 | 19 | −0.0101(1 ± 6) | −0.01105 | 29 | 0.0014(1 ± 10) | 0.00145 |
| 10 | 0.061(48 ± 10) | 0.06527 | 20 | −0.0102(8 ± 7) | −0.01110 | 30 | 0.0014(8 ± 9) | 0.00174 |

We move on to the projected structure factor $\bar{s}(k)$, shown in the same figure as $g(r)$. Also, here, there is a significant difference when $k \to 0$. For the fit, $\bar{s}_{\mathrm{fit}}(k)$ is positive for all $k$ whereas for the directly calculated coefficients, $\bar{s}(k)$ goes negative at around $k\ell = 0.3$ (see figure 15a) for a zoom in). At larger $k$, the two functions appear essentially the same. We see here that the fit is actually working better than the directly computed coefficients, in the sense that $s(k)$ should be positive for all $k$.

If we now scale away the Gaussian dampening and look at $P(k)$, in Figure 15b), we see that there is another discrepancy developing beyond $k\ell = 2$. This discrepancy, of course, has to do with the difference in the coefficients. However, we argue that its origin is also related to machine precision problems. We consider Figure 16 to shed some light on the phenomena. In the figure, we plot the $n$-dependence for $P(k)$. In the left column, the coefficients $c_n$ are used, whereas, in the right column, the $c_n^{\mathrm{fit}}$ coefficients are used.

The upper panels a) and b) show the same view as in figure 12a). The two images are qualitatively the same, and both show an onset of large positive and negative oscillations appearing for large $k$ when $n \approx 35$. For $P_n$, we argued earlier that this was due to the coefficients $c_n$ not being adequately resolved. While that is still true, the fact that we see the same phenomena also for $P_n^{\mathrm{fit}}$ means that the resolution of the coefficients is not the whole story.

We believe that what we observe here is a machine precision error. Recall that when computing $P(k)$, we use the expansion given in (30). This expansion, for order n, contains terms up to order $k^{2n}$. As a consequence, for $n = 35$, the largest order term at $k = 4$ (where the instability begins), is of size $4^{70} \approx 10^{42}$. Given that double precision decimals only can keep track of 16 digits, and that the $\lambda_t^{(n)}$ are alternating in sign, numerical instability is to be expected.

We now turn to the lower panels c) and d). These zoom in on smaller $k$ and use a larger $K_{\max}$. For the extracted coefficients $c_n$, we see that for small $n$, $P_n(k)$ sometimes goes negative, which was already discussed in the previous section. We also see that for $n > 70$, the lobes with negative $P_n(k)$ start reappearing with increased intensity. For the fitted coefficients $c_n^{\mathrm{fit}}$ d), the same phenomenon is observed at small $n$, but not at large $n$. For $n > 35$, $P_n(k)$ (with small $k$) is positive for all values of $n$ we have examined. This is a clear signal that the coefficients $c_n$ are not resolved with enough accuracy, whereas the fitted coefficients $c_n^{\mathrm{fit}}$ are well behaved.

Finally, we compute the magneto-roton gap for the fitted parameters. A comparison between $\Delta(k)$ and $\Delta^{\mathrm{fit}}(k)$ can be seen in figure 15b), for $K_{\max} = 39$. Here, larger differences can be seen; at large $k$, $F^{\mathrm{fit}}(k) < F(k)$, which stems from $P^{\mathrm{fit}}(k) < P(k)$. This results in a reduced $\Delta^{\mathrm{fit}}(k)$ compared with $\Delta(k)$. However, the difference between the two is not that dramatic, given that the fit only uses four parameters.

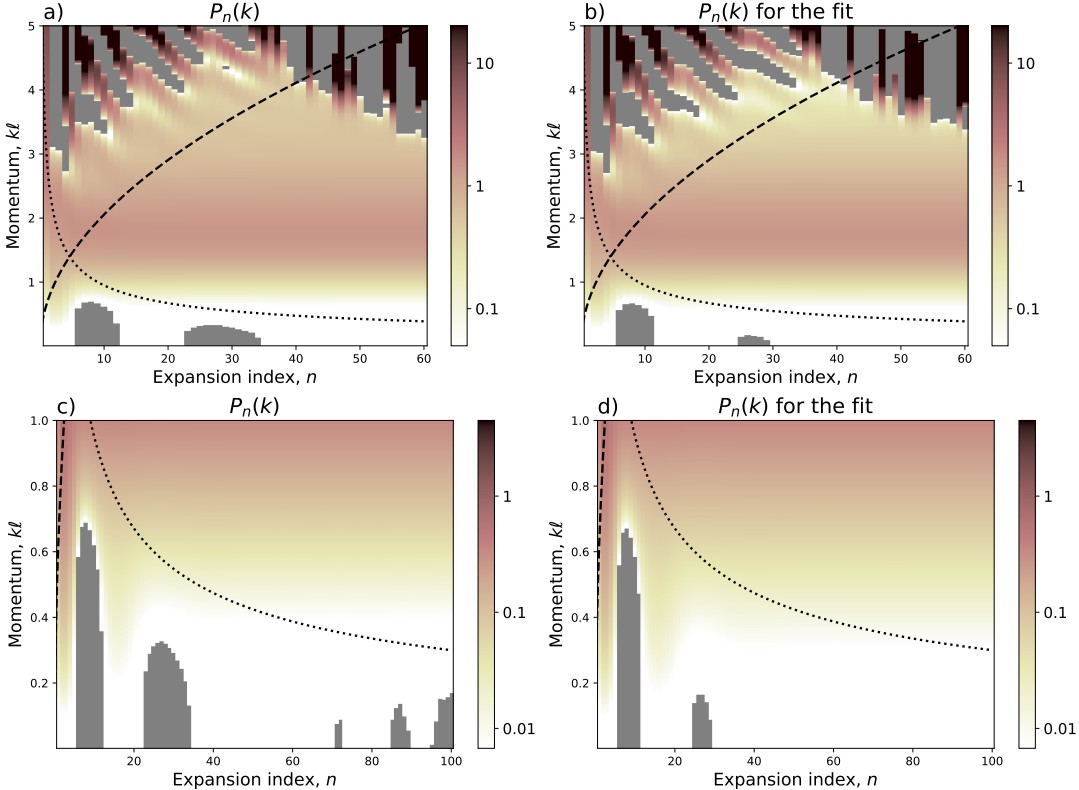

Figure 16: $n$-dependence for $P(k)$ for for $\nu = 1/3$. In a) and c), the coefficients $c_n$ are used, whereas in b) and d), the fit in equation (23) is used. In all plots, $P_n \leq 0$ is marked with gray. Panels a) and b) have the same view, which is also identical to that in figure 12a). The two images are qualitatively the same, and we infer that the appearance of (spurious) large positive and negative oscillations in $P_n(k)$ at large $k$ near $n = 35$ are due to machine precision errors, and not uncertainties on the coefficients $c_n$. The lower panels (c) and d) zoom in on smaller $k$ and uses larger $n$. We see there, that using the extracted coefficients c), $P_n \leq 0$ for larger $n$. This is a clear signal that the coefficients are $c_n$ are not resolved to enough accuracy. For the fitted coefficients in d), the same phenomena are not observed, and $P_n(k) > 0$ for all values of $k$.

To summarize, we have explored the possibility of directly using the four-parameter fit instead of the coefficients $c_n$. We find that these coefficients distort the $r \to 0$ region of $g(r)$, but do keep $\bar{s}(k) > 0$ when $k \to 0$. We also identified an important point for further improvement: the numerical instability in the large-$k$ expansion of $P(k)$.

Even though the final magneto-roton gap came out slightly differently, we find these results encouraging and worthy of further exploration. Further, we note that the fit results can be easily improved upon by *e.g.* only using the exact coefficients for low $n$ and then switching to the fit at high $n$.

# 8 Conclusions and Discussion

We have developed a basis for the expansion of FQH-pair correlation functions, which allows for a controlled extrapolation to the thermodynamic limit. This expansion should enable results involving pair correlations and quasihole densities to be more easily compared and reused

in later works.

We first reviewed the expansion due to Girvin and collaborators (cf. refs [4, 5]). This expansion is ill-suited for extrapolation to the thermodynamic limit since it is highly non-orthogonal. As a consequence, it renders causes the evaluation of the expansion coefficients to be numerically unstable. We have found that by orthogonalizing the original basis using the Gram-Schmidt procedure, a new basis is obtained in the form of (modified) Jacobi polynomials. Using this basis, we have been able to extract pair correlation expansion coefficients for a wide range of wavefunctions, including the Laughlin series, composite fermions with both reverse and direct flux-attachment, as well as Moore-Read and Bonderson-Slingerland states. We have also been able to extrapolate these coefficients to the thermodynamic limit. This way, we obtain expressions for the correlation functions on an infinite disk.

We further applied the expansion to both abelian and non-abelian quasiholes and found that the procedure works equally well. There are two minor complications. Firstly, a coefficient $c_0$ has to be introduced to allow for a nonzero density at the centre of the quasihole. This coefficient cannot be computed through an inner-product procedure and needs to be fitted for. Secondly, the one-particle correlation functions need many more samples than pair correlation functions to give the same degree of numerical convergence.

We showed a direct application of the expansion where we estimated the Magneto-Roton gap for all wave correlation functions where a thermodynamic scaling was performed. We could accurately construct the projected structure factor, and consequently the magneto-roton gap, in a window of $k_L \lesssim k \lesssim k_U$ where $k_L \propto 1/\sqrt{K_{\max}}$ and $k_U \propto \sqrt{K_{\max}}$.

The lack of precision for small $k$ is mainly related to the uncertainty in the polynomial $P(k)$ in the structure factor, and one may improve on this bound by taking more terms into account when computing $P(k)$ for small $k$. Such a scheme would be more advanced than that presented here, as it would require $K_{\max}$ to be $k$-dependent when computing $P(k)$, and was not attempted in this work. For larger $k$, one may also need to implement higher precision arithmetic or find a differential equation with $I_n(k)$ as its solution.

We demonstrated that the extrapolated coefficients $c_n$ show a lot of structure, and can be well approximated with $c_n \propto \exp(\tau\sqrt{n})\cos(\omega\sqrt{n})$, where $\tau$ and $\omega$ are real coefficients. The four-parameter expression works remarkably well, especially at large distances. Also, while four parameters suggest a lot of freedom, this is, in fact, just the decay length, frequency, amplitude, and phase of an oscillation. Further, the approximate parameter values can be read off directly from the graph and agree well with the parameters obtained by the fitting routine. By fitting $c_n$ to this form, one may construct an expression for $g(r)$ that effectively has $K_{\max} = \infty$. Such an expression can also be used as a starting point for a single-mode approximation that is well defined in the limit $k \to 0$. Identifying this simple parametrization has been one of the more exciting aspects of this work. These parameters could prove crucial in comparing and distinguishing candidate descriptions of FQH states at the same filling fraction.

The presented formalism can, with minor modifications, be generalized to, e.g., bosonic wave functions. The main difference would be that the correlation hole at $r = 0$ not necessarily is maximally excluding, i.e. $g(0) \neq 0$. At the pair correlation function level, it is an analogous phenomenon to the nonzero density that the non-abelian quasihole had, in the case of the fermionic Moore-Read state at $\nu = 1/2$. A nonzero $g(0)$ would be present for the $\nu = 1$ bosonic Moore-Read state, where two bosons (but not three) can be at the same position.

Another natural extension is to allow for non-isotropic wave functions. At the simplest level, this is done by adding an angular momentum $l$ phase factor $e^{il\theta}$ to the basis $G_n$. Allowing for an angular degree of freedom would allow us to make contact with the bi-metric [49, 51, 56, 63, 64] theory and the graviton mode. It is also a natural extension when considering correlation functions on the torus or strip/rectangle, where rotation symmetry is explicitly broken.

However, already with the current development of the correlation functions expansion and the available coefficients, many follow-up projects are immediately within reach. These include calculating interaction energies of quasiholes (assuming they don't deform too much in each other's presence) and different types of gap estimates.

The methods presented here still leave much room for improvement. In particular, in this work, the coefficients $c_n$ were computed using pre-binned approximations of $g(r)$. The pre-binning limits the resolution reachable in the coefficients $c_n$ for large $n$. The reason is that the bin sizes start becoming comparable with the wavelength of the basis functions, $G_n$. Thus, a better approach would be to compute $c_n$ directly using the Monte-Carlo data, with no intermediate pre-binning.

An interesting application of our method would be to extract the Fermi wave vector $k_F$ of composite fermions and compare it with experimental results [65]. Earlier attempts [66,67] to extract the $k_F$ of CFs from finite-size systems by fitting to $g(r)$ could get reasonable estimates of $k_F$ that were roughly consistent with experiments. The error bars were, however, still relatively large. Since our method can produce high-quality estimates of $g(r)$ in the thermodynamic limit, it is well suited for this problem.

## Acknowledgements

We would like to thank Hans Hansson, Steve Simon and Eddy Ardonne for fruitful discussions. This work was supported through SFI Principal Investigator Awards 12/IA/1697 and 16/IA/4524. This work is part of the D-ITP consortium, a program of the Netherlands Organisation for Scientific Research (NWO) that is funded by the Dutch Ministry of Education, Culture and Science (OCW). We also wish to acknowledge the SFI/HEA Irish Centre for High-End Computing(ICHEC) for the provision of computational facilities and support through project nmphy011b and nmphy013b. We are grateful for the use of the code package Hammer, which was used for the numerical simulations.

## A  Spherical pair correlation function for $\nu = 1$

In this section we derive the pair correlation function for the $\nu = 1$ state. In general, when the wavefunction in question is a single determinant, we have $\langle \mathbf{r}_j | \Psi \rangle = \Psi(\mathbf{r}_j) = \frac{1}{\sqrt{N!}} \text{Det}\left[\phi_i(\mathbf{r}_j)\right]$ where $\phi_i$ are the occupied single particle orbitals. Removing state number $k$, in this case the one corresponding to $\mathbf{r}_1$, gives

$$\langle \mathbf{r}_{j>1} | a_k | \Psi \rangle = \frac{1}{\sqrt{N-1}} \text{Det}\left[\phi_{i \neq k}(\mathbf{r}_{j>1})\right], \tag{A.1}$$

where $a_k$ is the annihilation operator.

Using the expansion of a determinant into its minors

$$\text{Det}[a_{ij}] = \sum_{k=1}^{n} (-1)^{k+l} a_{kl} \text{Det}[a_{i \neq k, j \neq l}],$$

we first find an expression for the expectation value of the density:

$$\rho(\mathbf{r}_1) = N \int \prod_{i>1} dS_i \, |\Psi|^2 = \sum_{k=1}^{N} |\phi_k(\mathbf{r}_1)|^2, \tag{A.2}$$

where we also have used that $\int \prod_{j>1} dS_j \; |r_{j>1}\rangle\langle r_{j>1}|$ is an identity in the space of $N-1$ particles and that $\Psi$ with one state removed yields an orthonormal set: $\langle\Psi|a_k^\dagger a_l|\Psi\rangle = \delta_{kl}$.

In a similar manner we can find an expression for the pair correlation function:

$$
\begin{aligned}
g\big(|r_1 - r_2|\big) &= \frac{N(N-1)}{\rho^2} \int \prod_{i>2} dS_i \; |\Psi|^2 \\
&= \frac{1}{\rho^2} \sum_{k,l=1}^{N} \Big( |\phi_k(r_1)|^2 |\phi_l(r_2)|^2 - \phi_k^*(r_1)\phi_l(r_1)\phi_l^*(r_2)\phi_k(r_2) \Big).
\end{aligned} \tag{A.3}
$$

At this point we turn to the state $\nu = 1$, *i.e.* a determinant consisting of all the lowest Landau level functions for the chosen magnetic flux. It is convenient to use spinor coordinates

$$
u = \cos(\theta/2)\exp(i\phi/2), \quad v = \sin(\theta/2)\exp(-i\phi/2), \tag{A.4}
$$

with the polar and azimuthal angles $(\theta, \phi)$. The spherical lowest Landau level single particle wavefunctions are given as [68]

$$
\phi_k(u,v) = \sqrt{\frac{2Q+1}{4\pi Q}\binom{2Q}{k}}(-1)^k u^k v^{2Q-k}, \tag{A.5}
$$

with $k \in \{0,1,\ldots,2Q\}$ and in terms of the spinor coordinates in (A.4). As a consistency check we find using (A.2) that the density is $\rho = N/A$. Following (A.3) we get for the pair correlation function, in terms of the chord distance $r = 2R|u_1 v_2 - u_2 v_1|$,

$$
g_1(r) = 1 - \Big(1 - |u_1 v_2 - u_2 v_1|^2\Big)^{2Q} = 1 - \Big(1 - \frac{r^2/2}{2Q}\Big)^{2Q}. \tag{A.6}
$$

In terms of the unit distance (11) we have $g_1(\eta) = 1 - (1-\eta^2)^{2Q}$. In the limit of infinite radius we regain the planar $\nu = 1$ function (6):

$$
\lim_{2Q \to \infty} g_1(r) = 1 - e^{-r^2/2}. \tag{A.7}
$$

Note that expressions for the pair correlation functions of all the excited states of $\nu = 1$ can be obtained in the same manner, due to the fact that they all consist of a single Slater determinant.

## B  The Spherical expansion basis

We now derive the expansion basis for correlation functions on the sphere. The derivation here closely follows the one for the plane in Ref. [5]. As a starting point we write the wavefunction in a form exposing the dependence on particle 1 and 2:

$$
\Psi = \sum_{\substack{j,k \\ j<k}}^{2Q} a_{jk}(u_3,v_3,\ldots,u_N,v_N)\big(\phi_j(u_1,v_1)\phi_k(u_2,v_2) - \phi_k(u_1,v_1)\phi_j(u_2,v_2)\big), \tag{B.1}
$$

where the antisymmetry of $\Psi$ under exchange of $(u_1,v_1)$ and $(u_2,v_2)$ is explicit and $a_{jk} \in \mathbb{C}$. Using that the state is isotropic we can assume that particle 1 is at the north pole without loss of generality. With the distance between the particles measured in unit length $\eta = \frac{r}{2R}$ we then have for the spinor coordinates:

$$
\begin{aligned}
(u_1,v_1) &= (1,0), \\
(u_2,v_2) &= \big(\sqrt{1-\eta^2}e^{i\phi_2/2}, \eta e^{-i\phi_2/2}\big),
\end{aligned} \tag{B.2}
$$

where $\phi_2$ is the azimuthal coordinate of the second particle. Using this together with (A.5), the first term in the brackets of (B.1) is zero unless $j = 2Q$, while the same holds true for the second term with $k = 2Q$. Since $j < k \le 2Q$ the first term vanishes, and we end up with

$$\Psi = -\sum_{j=0}^{2Q-1} a_{j,2Q}(1-\eta^2)^{\frac{j}{2}}\eta^{2Q-j}e^{i(j-Q)\phi_2}. \tag{B.3}$$

Substituting this into (2) and using the fact that $g(\eta)$ should be independent of $\phi_2$ then yields

$$g(\eta) = \sum_{k=0}^{2Q-1} A_k(1-\eta^2)^k\eta^{4Q-2k}, \tag{B.4}$$

where $A_k = \frac{N(N-1)}{\rho^2}\int\prod_{i>2}d\Omega_i \, |a_{k,2Q}|^2$. In order to extract the terms of $g_1$ (12) we define expansion coefficients by $A_k = \binom{2Q}{k} + d_k$. After reordering the terms by $k \to 2Q-k$ so that the functions with low indices are centered around the north pole, we end up with

$$g(\eta) = 1 - \left(1-\eta^2\right)^{2Q} + \sum_{k=1}^{2Q} d_k f_k(\eta),$$
$$f_k(\eta) = (1-\eta^2)^{2Q-k}\eta^{2k}. \tag{B.5}$$

This constitutes a spherical expansion of the pair correlation function.

## C   Orthonormality of the spherical basis in eq. (16)

In the main text it is mentioned that we use the Gram-Schmidt orthogonalization procedure to find the basis in equation (16). Rather than giving that argument, we find it more illuminating to prove that the basis is orthonormal. Combined with the observation that $G_n$ is a linear combination of $f_1, \ldots, f_n$ it should be clear that $G_n$ can be reached though the Gram-Schmidt procedure.

We thus set out to prove that the following functions are orthonormal under the integration measure $dS = 8\pi Q\eta d\eta$:

$$G_n(\eta) = \mathcal{N}_n\eta^2(1-\eta^2)^{2Q-n}J_{n-1}^{(2,4Q+1-2n)}(1-2\eta^2),$$
$$\mathcal{N}_n = \sqrt{\frac{(4Q+2-n)(4Q+1-n)(4Q-2n+1)}{4\pi Qn(n+1)}}, \tag{C.1}$$

where $1 \le n \le 2Q$.

### C.1   Orthogonality

We begin by showing that (C.1) is an orthogonal set. For this the normalization is irrelevant, and we ignore all constants. Note that although the inner product $\langle G_n, G_m \rangle$ is reminiscent of that in the orthogonality relation between two Jacobi polynomials $J_k^{(\alpha,\beta)}$ [22] we cannot use this relation directly. This is because the relation assumes that the parameters $(\alpha, \beta)$ are equal in the two polynomials, which is not the case for $G_n$ and $G_m$ when $n \ne m$.

As a first step we substitute the variable $x = 1-2\eta^2$ for $\eta$. This leads to $dS = -2\pi Q dx$ and gives the integration limits $x(\eta = 0) = 1$ and $x(\eta = 1) = -1$. Thus (C.1) gives the following inner product:

$$\langle G_n, G_m \rangle \propto \int_{-1}^{1} dx \left(\frac{1-x}{2}\right)^2 \left(\frac{1+x}{2}\right)^{4Q-n-m} J_{n-1}^{(2,4Q+1-2n)}(x)J_{m-1}^{(2,4Q+1-2m)}(x). \tag{C.2}$$

At this point it is convenient to introduce the shorthand

$$
\begin{aligned}
A(x) &= 1 - x\,, \\
B(x) &= 1 + x\,.
\end{aligned}
\tag{C.3}
$$

We note that $A(-1)B(-1) = A(1)B(1) = 0$. With this convention Rodrigues' formula [22] reads

$$
J_k^{(\alpha,\beta)}(x) = \frac{(-1)^k}{2^k k!} A^{-\alpha} B^{-\beta} \frac{d^k}{dx^k}\left(A^{\alpha+k} B^{\beta+k}\right).
\tag{C.4}
$$

Using this (C.2) can be written as

$$
\langle G_n, G_m \rangle \propto \int_{-1}^{1} dx\, A^{-2} B^{n+m-2-4Q} \frac{d^{n-1}}{dx^{n-1}}\left(A^{n+1} B^{4Q-n}\right) \frac{d^{m-1}}{dx^{m-1}}\left(A^{m+1} B^{4Q-m}\right).
\tag{C.5}
$$

We will show that this equals zero when $n \neq m$ using repeated integration by parts. In preparation we observe that

$$
\text{The polynomial } \frac{d^k}{dx^k}\left(A^p B^q\right),\ \text{ has a factor } AB\ \text{ when }\ p > k < q\,.
\tag{C.6}
$$

Without loss of generality we assume $n < m$. A first integration by parts leaves (C.5) as

$$
\begin{aligned}
\langle G_n, G_m \rangle \propto & \left[ \left\{ A^{-2} B^{n+m-2-4Q} \frac{d^{n-1}}{dx^{n-1}}\left(A^{n+1} B^{4Q-n}\right) \right\} \left\{ \frac{d^{m-2}}{dx^{m-2}}\left(A^{m+1} B^{4Q-m}\right) \right\} \right]_{-1}^{1} \\
& - \int_{-1}^{1} dx\, \frac{d}{dx}\left\{ A^{-2} B^{n+m-2-4Q} \frac{d^{n-1}}{dx^{n-1}}\left(A^{n+1} B^{4Q-n}\right) \right\} \frac{d^{m-2}}{dx^{m-2}}\left(A^{m+1} B^{4Q-m}\right).
\end{aligned}
\tag{C.7}
$$

First we show that the boundary term is zero. We note that the first factor $A^{-2} B^{n+m-4Q} \frac{d^{n-1}}{dx^{n-1}}\left(A^{n+1} B^{4Q-n}\right)$ is zero or a polynomial of order $m-2$, and therefor regular.

Next we examine the second factor: $\frac{d^{m-2}}{dx^{m-2}}\left(A^{m+1} B^{4Q-m}\right)$. Looking at the derivative and the polynomial powers we have that $m-2 < m+1$ and that $m-2 < 4Q-m$ (since $m \leq 2Q$). (C.6) therefore implies that it has a factor $AB$. Thus the boundary term is a product of regular terms and a factor $AB$, and therefore equals zero when evaluated at both boundaries $x = -1$ and $x = 1$.

Applying further integrations by parts will produce boundary terms similar to that in (C.7) but with derivatives acting on the whole of the first factor, in increasing order, while the derivative in the second factor decreases in order. This does not change the reasoning in the previous paragraph, and we see that all boundary terms vanish. Thus the result after $k$ integrations by parts is

$$
\langle G_n, G_m \rangle \propto \int_{-1}^{1} dx\, \frac{d^k}{dx^k}\left\{ A^{-2} B^{n+m-2-4Q} \frac{d^{n-1}}{dx^{n-1}}\left(A^{n+1} B^{4Q-n}\right) \right\} \frac{d^{m-1-k}}{dx^{m-1-k}}\left(A^{m+1} B^{4Q-m}\right).
\tag{C.8}
$$

We see that the first factor in (C.8) will have order zero, *i.e.* be a constant, when $k = m-2$. At this point the integrand is a pure differential:

$$
\langle G_n, G_m \rangle \propto \int_{-1}^{1} dx\, \frac{d}{dx}\left(A^{m+1} B^{4Q-m}\right) = \left[ A^{m+1} B^{4Q-m} \right]_{-1}^{1} = 0\,,
\tag{C.9}
$$

concluding our proof of orthogonality.

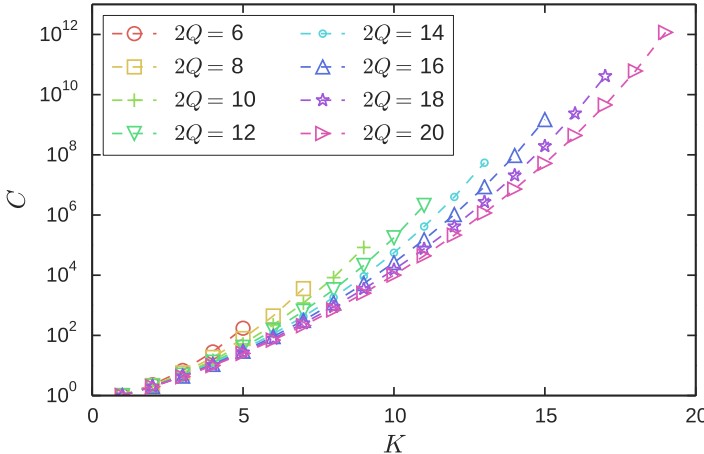

Figure 17: Condition number of the matrix in (D.2) at chosen values of $2Q$, plotted against dimension $K_{\text{max}}$.

## C.2   Orthonormality

To prove that the functions are orthonormal it only remains to show that $\langle G_n, G_n \rangle = 1$. In this case the caveat of $\alpha \neq \beta$ no longer holds, and we may use the Jacobi polynomial orthogonality relation directly, since the two functions now have the same parameters. We remind that the Jacobi polynomial orthogonality relation reads

$$\int_{-1}^{1} (1-x)^{\alpha}(1+x)^{\beta} J_m^{(\alpha,\beta)}(x) J_n^{(\alpha,\beta)}(x) = \frac{2^{\alpha+\beta+1}}{2n+\alpha+\beta+1} \frac{\Gamma(n+\alpha+1)\Gamma(n+\beta+1)}{\Gamma(n+\alpha+\beta+1)n!}. \quad \text{(C.10)}$$

Comparing (C.10) with the integral $\langle G_n, G_n \rangle$ we find that $\alpha = 2$, $\beta = 4Q+1-2n$, $m = n$ precisely gives (C.5). This proves that $G_n$ is normalized and that the set of functions $G_n$ are orthonormal.

## D   Condition number

In the main text we argued that the orthogonal spherical basis is stable whereas the non-orthogonal one is not. We here present a quantitative argument as to why this is the case based of the condition number. The condition number $C$ gauges the stability of a map between two quantities: if it is big it means that a small change in one induces a large change in the other. As a rule of thumb, if $C \sim 10^k$, up to $k$ digits of accuracy may be lost in the map [69].

For a linear transformation $\boldsymbol{a} \mapsto M\boldsymbol{b}$ the condition number is defined as

$$C_M = ||M|| \cdot ||M^{-1}||, \quad \text{(D.1)}$$

where in our case we use the Euclidian norm. Transforming between the two spherical bases involves the Gram matrix $M_{nk}$ defined through $G_n(\eta) = \sum_{k=1}^{n} M_{nk} f_k(\eta)$, and from (13) and (16) this is given as

$$M_{nk} = \mathcal{N}_n \sqrt{\frac{8\pi(4Q-2n)!(2n-1)}{4Q+1} \frac{(2+n-1)!}{(4Q+2-n)!(n-1)!}} \binom{n-1}{k-1} \frac{(4Q+1-n-k)!}{(k+1)!}. \quad \text{(D.2)}$$

Figure 17 plots $C_M$ for this matrix at some chosen values of the flux $2Q$ against the number of functions included $K_{\text{max}}$, *i.e.* the dimension of $M$. For a given flux the condition number

grows faster than exponentially with the dimension, quickly becoming very large. This indicates that the non-orthogonal coefficients will be imprecise while the orthogonal remain accurate.

## E   Evaluating $G_n(\eta)$ as a differential equation

In the main text we mentioned that equation (17), albeit being exact, is not easy to evaluate to sufficient accuracy. Instead we will in this appendix derive a (stable) second order differential equation that we can solve numerically.

From (16) we know that we may write

$$G_n(\eta) = \mathcal{N}_n \eta^2 \left(1 - \eta^2\right)^{2Q-n} J_{n-1}^{(2,4Q-2n+1)},$$

where $J_{n'}^{\alpha,\beta}$ is a Jacobi polynomial.

The Jacobi polynomials $y(x) = J_{n'}^{(\alpha,\beta)}(x)$ solves the differential equation

$$y'' = Ay' + By \tag{E.1}$$

for the parameters

$$A = \frac{\beta - \alpha - (\alpha + \beta + 2)x}{x^2 - 1},$$

$$B = \frac{n'\left(n' + \alpha + \beta + 1\right)}{x^2 - 1}.$$

We now define the function $f(x) = g(x) \cdot y(x)$ where $g(x) = \left(1 - \eta^2\right)^M = \left(\frac{x+1}{2}\right)^M$. We then have $G_n(\eta) = \mathcal{N}_n \eta^2 f(\eta)$, with $M = 2Q - n$, $\alpha = 2$, $\beta = 4Q - 2n + 1$ and $n' = n - 1$.

We now derive a differential equation for $f$. By taking successive derivatives of $f$ and and using equation (E.1) we arrive at

$$f'' = \left[B + \frac{g''}{g} - \left(2\frac{g'}{g} + A\right)\left(\frac{g'}{g}\right)\right]f + \left(A + 2\frac{g'}{g}\right)f'.$$

In our case $g = \left(\frac{x+1}{2}\right)^M$ so $\frac{g'}{g} = \frac{M}{x+1}$ and $\frac{g''}{g} = \frac{M(M-1)}{(x+1)^2}$. This gives the second order differential equation

$$f'' = \left(B - \frac{M}{x+1}\left(\frac{M+1}{x+1} + A\right)\right)f + \left(A + \frac{2M}{x+1}\right)f',$$

which may integrated numerically by your favorite method. We mention in this regard that the numerical integration is well behaved for the full length of the oscillatory part of $f$ but will diverge just beyond that point, due to the other solution growing exponentially fast. This is not so bad because the desired function $f$ is exponentially suppressed and can be safely set to zero by an appropriate algorithm whenever the numerical instability kicks in.

## F   The $\lambda_t^{(n)}$ coefficient

In this appendix we will work out the coefficients $\lambda_t^{(n)}$ appearing in equation (29) in the main text. Thus, we wish to compute the integral

$$I_n = (-1)^n \int_0^\infty dr \frac{r^3 e^{-\frac{1}{2}r^2}}{\sqrt{\pi n(n+1)}} J_0(kr) L_{n-1}^{(2)}\left(r^2\right),$$

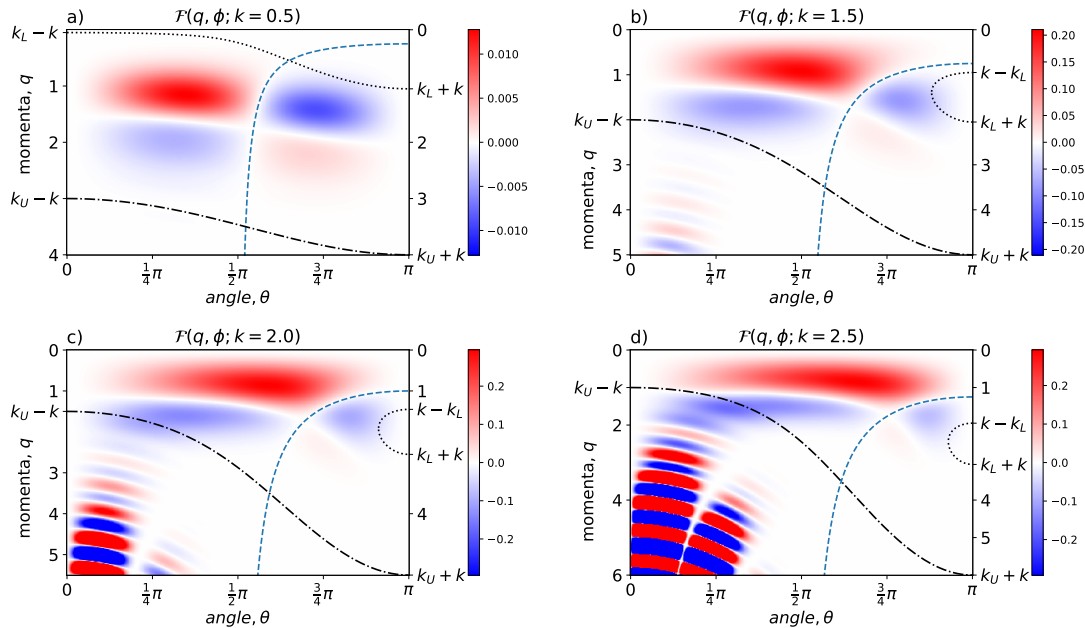

Figure 18: Inspection of how imposing a cutoff impedes on the gap integral (33), exemplified for the $\nu = 1/3$ Laughlin state, with $K_{\max} = 30$. The panels show $\mathcal{F}(q, \phi; k)$ with $k$ fixed as a) $k = 0.5$, b) $k = 1.5$, c) $k = 2.0$, d) $k = 2.5$. We here estimate $k_U = 3.5$ and $k_L = 0.55$, and a dashed (dotted) line marks when $|\vec{k} + \vec{q}| = k_U$ ($k_L$). a) For small $k$, the integrand is well behaved, and we may cut the integration off at $|\vec{k} + \vec{q}| = k_U$. b-c) For larger $k$, numerical noise, coming from badly resolved pars of $s(k)$ start creeping in from larger $q$. d) If $k$ is too large, the numerical noise merges with the converged parts of the integral and pollutes the result.

and expand it in powers of $k^2$. The first step is to use that the Laguerre polynomials have an expansion

$$L_{n-1}^{(2)}\left(r^2\right) = \sum_{s=0}^{n-1} (-1)^s \binom{n+1}{n-1-s} \frac{r^{2s}}{s!}.$$

Inserting this expansion yields the expression,

$$I_n = \frac{(-1)^n}{\sqrt{\pi n(n+1)}} \sum_{s=0}^{n-1} (-1)^s \binom{n+1}{n-1-s} \frac{1}{s!} \overbrace{\int_0^\infty dr e^{-\frac{1}{2}r^2} J_0(kr) r^{2(s+1)+1}}^{S_{s+1}(k)}, \qquad \text{(F.1)}$$

where we have also defined

$$S_n(k) = \int_0^\infty dr\, r^{2n+1} e^{-\frac{1}{2}r^2} J_0(kr). \qquad \text{(F.2)}$$

The integral is evaluated by expanding the Bessel function as

$$J_0(kr) = \sum_{m=0}^\infty \frac{(-1)^m}{m!\Gamma(m+1)} \left(\frac{kr}{2}\right)^{2m}, \qquad \text{(F.3)}$$

and integrating the resulting Gaussian integral. Insertion of (F.3) into (F.2) gives

$$S_n = \sum_{m=0}^{\infty} \frac{(-1)^m}{(m!)^2} \left(\frac{k}{2}\right)^{2m} \int_0^{\infty} dr\, r^{2(m+n)+1} e^{-\frac{1}{2}r^2},$$

which after using the following standard formula for Gaussian integration

$$\int_0^{\infty} x^{2n+1} e^{-\frac{x^2}{a^2}}\, dx = \frac{n!}{2} a^{2n+2},$$

gives

$$S_n = 2^n \sum_{m=0}^{\infty} \frac{(m+n)!}{(m!)^2} \left(-\frac{k^2}{2}\right)^m.$$

We proceed by renaming $x = k^2$ and rewrite $\frac{(m+n)!}{m!} x^m = \frac{d^n}{dx^n} x^{m+n}$. We may then perform the sum over $m$ and obtain

$$S_n(x) = 2^n \frac{d^n}{dx^n} x^n e^{-\frac{x}{2}}.$$

We then apply the binomial rule for derivatives

$$\frac{d^n}{dx^n} g(x) f(x) = \sum_{t=0}^{n} \binom{n}{t} g^{(n-t)}(x) f^{(t)}(x),$$

to $S_n(x)$ with $g = x^n$ and $f = e^{-\frac{x}{2}}$ we have $f^{(t)}(x) = \left(-\frac{1}{2}\right)^t e^{-\frac{x}{2}}$ and $g^{(k)}(x) = \frac{n!}{(n-k)!} x^{n-k}$. Putting all the pieces together gives

$$S_n = e^{-\frac{x}{2}} 2^n \sum_{t=0}^{n} \binom{n}{t} \frac{n!}{t!} \left(-\frac{x}{2}\right)^t, \tag{F.4}$$

which is a power series expansion in $x = k^2$.

Next, we reinsert (F.4) into (F.1) and obtain

$$I_n = \frac{-e^{-\frac{x}{2}}(-1)^n}{\sqrt{\pi n(n+1)}} \sum_{s=1}^{n} (-1)^s \binom{n+1}{n-s} 2^s \sum_{t=0}^{s} \binom{n}{t} \frac{s}{t!} \left(-\frac{x}{2}\right)^t, \tag{F.5}$$

where we also shifted the sum over $s$ by one. We now would like to make the identification

$$I_n = -e^{-\frac{x}{2}} \sum_{t=0}^{n} \lambda_t^{(n)} x^t.$$

To be able to perform the identification we change the order of the sums in (F.5) from $\sum_{s=1}^{n} \sum_{t=0}^{s}$ to $\sum_{t=0}^{n} \sum_{s=t}^{n}$. The new sum formally has an extra term with $t = s = 0$, which is zero. Rearranging (F.5) then leads to

$$I_n = \frac{-e^{-\frac{x}{2}}(-1)^n}{\sqrt{\pi n(n+1)}} \sum_{t=0}^{n} \sum_{s=t}^{n} (-1)^s \binom{n+1}{n-s} 2^s \binom{n}{t} \frac{s}{t!} \left(-\frac{x}{2}\right)^t, \tag{F.6}$$

where we now identify $\lambda_t^{(n)}$ as

$$\lambda_t^{(n)} = \frac{(-1)^n}{t!\sqrt{\pi n(n+1)}} \sum_{s=t}^{n} (-2)^{s-t} \binom{n+1}{n-s} \binom{n}{t} s, \tag{F.7}$$

which is equation (29) in the main text.

## G   Cutoff in the integral (33)

In the main text, it was noted that we do not trust $P(k)$ for all values of $k$, but only in the range $k_L \lesssim k \lesssim k_U$. This uncertainty spills over to the integral (33) such that we naively would expect that we should not integrate further than $q < k_U - k$. This means that our estimate of $F(k)$ is only valid in the range $k \lesssim k_U/2$.

To see how this comes about, we write the integral in (33) as

$$F(k) = \int_0^\pi d\phi \int_0^\infty dq \, \mathcal{F}(q, \phi; k),$$

and examine $\mathcal{F}(q, \phi; k)$ further. This is done in figure 18 for the $\nu = 1/3$ Laughlin state, with $K_{\max} = 30$ and $k = 0.5, 1.5, 2.0, 2.5$. For this number of terms we estimate $k_U = 3.5$ and $k_L = 0.55$. In these figures $q$ is plotted in the range $0 < q < k_U + k$ and a dashed (dotted) line marks when $|\vec{k} + \vec{q}| = k_U$ ($k_L$). For small values of $k$ (figure 18a) nothing is really going on close to $|\vec{k} + \vec{q}| = k_U$, but as $k$ grows (figure 18b and figure 18c), one can see how large divergent contributions are creeping closer to $|\vec{k} + \vec{q}| = k_U$ from larger $q$.

When $k$ is too large (figure 18d) the regions with converged contributions to $F(k)$ merge with the unconverged regions. The merges takes place precisely where $|\vec{k} + \vec{q}| = k_U$. In this work we regularize the integral by only integrating up to $|\vec{k} + \vec{q}| \leq k_U$ and discarding the rest of the integral. While this gives excellent results when $k$ is sufficiently small, the precise value of the integral will depend on $k_U$ when $k \approx k_U/2$, which is seen in figures 18c and 18d. By varying $k_U$ we get an estimate of how sensitive $F(k)$ is to the cutoff and thus for how large values of $k$ we may trust the calculation.

## H   Extra Data

In this section we list some auxiliary data that was not explicitly mentioned in the main text. We list

- The pair correlation functions for reverse flux composite fermions at $\nu = 2/3$ and $\nu = 3/5$; Figure 19.

- Extensive list of expansion coefficients the Laughlin $\nu = 1/5$ and $\nu = 1/7$ states in Table 7.

- Expansion coefficients CF $\nu = 2/5$ and $\nu = 3/7$ states as well as the reverse flux state at $\nu = 2/3$ and $\nu = 3/5$ in Table 8.

- Expansion coefficients for the Moore-Read wavefunction at filling $\nu = 2 + 1/2$ and the Reverse flux modified Laughlin wavefunction at $\nu = 1/3$ in Table 9

- The expansion coefficients (including their uncertainties) for all the states treated in this work is available in digital form in the supplementary material.



Table 7: Expansion coefficients $c_n$ for the Laughlin $\nu = 1/5$ and $\nu = 1/7$ states.

| $n$ | $\nu=1/5$ | $\nu=1/7$ | $n$ | $\nu=1/5$ | $\nu=1/7$ | $n$ | $\nu=1/5$ | $\nu=1/7$ | $n$ | $\nu=1/7$ |
|---|---|---|---|---|---|---|---|---|---|---|
| 1 | 3.5613(97±12) | 3.73192(5±5) | 36 | −0.0382(7±8) | −0.043(50±17) | 71 | −0.005(28±13) | 0.046(69±15) | 106 | −0.013(46±12) |
| 2 | 2.8439(4±3) | 3.4647(95±19) | 37 | −0.030(99±16) | −0.071(59±12) | 72 | −0.005(25±12) | 0.044(14±12) | 107 | −0.012(50±18) |
| 3 | 1.8254(8±4) | 3.0906(9±3) | 38 | −0.023(44±11) | −0.094(50±18) | 73 | −0.0050(2±10) | 0.041(03±18) | 108 | −0.010(91±19) |
| 4 | 0.7141(2±4) | 2.4537(1±3) | 39 | −0.016(09±13) | −0.112(40±15) | 74 | −0.004(56±20) | 0.038(21±19) | 109 | −0.010(52±15) |
| 5 | −0.1679(0±5) | 1.5920(8±5) | 40 | −0.0097(7±7) | −0.125(60±19) | 75 | −0.004(24±15) | 0.034(3±3) | 110 | −0.009(71±13) |
| 6 | −0.6854(6±7) | 0.6622(1±6) | 41 | −0.003(86±13) | −0.135(05±10) | 76 | −0.004(26±11) | 0.030(79±16) | 111 | −0.008(08±16) |
| 7 | −0.8784(1±4) | −0.1632(7±6) | 42 | 0.001(95±12) | −0.140(33±11) | 77 | −0.0033(0±9) | 0.027(5±2) | 112 | −0.006(88±17) |
| 8 | −0.8504(3±7) | −0.7750(8±10) | 43 | 0.006(30±10) | −0.141(23±11) | 78 | −0.003(05±15) | 0.023(52±18) | 113 | −0.005(7±2) |
| 9 | −0.7015(7±7) | −1.1434(5±9) | 44 | 0.010(13±12) | −0.139(46±16) | 79 | −0.002(76±18) | 0.020(13±15) | 114 | −0.004(5±3) |
| 10 | −0.5051(7±9) | −1.2941(4±5) | 45 | 0.0127(2±9) | −0.135(19±17) | 80 | −0.002(05±18) | 0.016(09±19) | 115 | −0.003(6±2) |
| 11 | −0.306(69±10) | −1.278(43±14) | 46 | 0.014(44±15) | −0.128(3±3) | 81 | −0.001(89±16) | 0.012(50±19) | 116 | −0.002(74±14) |
| 12 | −0.1317(2±9) | −1.150(80±16) | 47 | 0.016(49±12) | −0.119(21±13) | 82 | −0.001(41±13) | 0.008(94±18) | 117 | −0.001(82±14) |
| 13 | 0.0079(5±9) | −0.9593(9±7) | 48 | 0.0170(6±9) | −0.108(5±3) | 83 | −0.001(03±15) | 0.005(52±16) | 118 | −0.000(8±2) |
| 14 | 0.110(52±13) | −0.740(17±13) | 49 | 0.017(11±13) | −0.097(2±2) | 84 | −0.000(80±14) | 0.002(5±2) | 119 | 0.000(7±2) |
| 15 | 0.1778(7±8) | −0.518(20±11) | 50 | 0.017(01±15) | −0.085(11±15) | 85 | −0.000(31±14) | −0.000(55±17) | 120 | 0.001(47±18) |
| 16 | 0.214(22±10) | −0.309(31±10) | 51 | 0.016(41±12) | −0.071(8±2) | 86 | −0.0002(7±10) | −0.003(33±18) | 121 | |
| 17 | 0.2255(5±9) | −0.123(5±2) | 52 | 0.015(16±15) | −0.058(90±14) | 87 | −0.000(24±14) | −0.005(61±20) | 122 | |
| 18 | 0.2181(3±10) | 0.033(94±12) | 53 | 0.013(96±15) | −0.046(5±2) | 88 | 0.0001(9±9) | −0.008(27±19) | 123 | |
| 19 | 0.197(27±11) | 0.161(95±15) | 54 | 0.012(67±16) | −0.033(6±2) | 89 | 0.000(6±2) | −0.010(72±15) | 124 | |
| 20 | 0.1676(8±7) | 0.260(29±14) | 55 | 0.010(98±15) | −0.021(6±2) | 90 | 0.000(57±15) | −0.012(31±13) | 125 | |
| 21 | 0.133(90±12) | 0.331(38±14) | 56 | 0.009(03±15) | −0.010(1±3) | 91 | 0.000(88±13) | −0.013(66±19) | 126 | |
| 22 | 0.0992(2±10) | 0.377(61±19) | 57 | 0.007(73±11) | 0.000(3±2) | 92 | 0.001(27±14) | −0.015(28±15) | 127 | |
| 23 | 0.065(54±11) | 0.401(16±14) | 58 | 0.006(16±12) | 0.009(54±17) | 93 | 0.001(30±13) | −0.016(6±2) | 128 | |
| 24 | 0.033(78±11) | 0.405(72±15) | 59 | 0.004(56±19) | 0.018(51±15) | 94 | 0.001(54±17) | −0.017(3±2) | 129 | |
| 25 | 0.0071(0±10) | 0.395(04±11) | 60 | 0.002(60±15) | 0.026(14±16) | 95 | 0.001(85±17) | −0.017(75±17) | 130 | |
| 26 | −0.0151(0±9) | 0.371(54±14) | 61 | 0.000(75±15) | 0.032(29±16) | 96 | 0.001(28±12) | −0.018(35±12) | 131 | |
| 27 | −0.0331(2±6) | 0.337(99±15) | 62 | 0.000(23±20) | 0.037(85±16) | 97 | 0.001(6±2) | −0.017(88±18) | 132 | |
| 28 | −0.0460(9±9) | 0.297(40±15) | 63 | −0.001(24±16) | 0.041(9±3) | 98 | 0.001(41±17) | −0.018(1±2) | 133 | |
| 29 | −0.0547(8±9) | 0.252(43±16) | 64 | −0.002(19±14) | 0.045(95±15) | 99 | 0.001(0±2) | −0.018(6±2) | 134 | |
| 30 | −0.059(98±12) | 0.205(45±12) | 65 | −0.0026(1±10) | 0.048(36±17) | 100 | 0.001(69±16) | −0.017(5±2) | 135 | |
| 31 | −0.061(61±13) | 0.157(78±12) | 66 | −0.003(60±12) | 0.049(81±11) | 101 | 0.001(60±15) | −0.017(35±19) | 136 | |
| 32 | −0.0601(9±9) | 0.111(15±13) | 67 | −0.004(05±17) | 0.050(4±2) | 102 | 0.001(0±3) | −0.017(5±2) | 137 | |
| 33 | −0.056(41±11) | 0.066(92±13) | 68 | −0.0046(6±10) | 0.050(2±2) | 103 | 0.000(91±14) | −0.016(5±2) | 138 | |
| 34 | −0.0512(1±10) | 0.025(68±15) | 69 | −0.004(88±11) | 0.049(61±15) | 104 | 0.000(69±19) | −0.015(75±16) | 139 | |
| 35 | −0.045(06±15) | −0.011(19±16) | 70 | −0.004(95±12) | 0.048(2±2) | 105 | 0.001(0±2) | −0.014(6±2) | 140 | |

Table 8: The first 20 expansion coefficients $c_n$ for the BS state at $\nu = 2 + 2/5$, the CF states at $\nu = 2/5$ and $\nu = 3/7$, as well as the reverse flux composite fermion states at $\nu = 2/3$ and $\nu = 3/5$.

| $n$ | BS $\nu=2/5$ | CF $\nu=2/5$ | CF $\nu=3/7$ | CF $\nu=2/3$ | CF $\nu=3/5$ |
|---|---|---|---|---|---|
| 1 | 1.53(2±5) | 2.01(7±3) | 1.79(9±2) | 0.651(63±15) | 0.891(7±6) |
| 2 | 1.0(04±10) | 0.63(18±17) | 0.54(04±16) | 0.240(6±3) | 0.271(6±2) |
| 3 | 0.49(3±7) | −0.19(2±3) | −0.19(3±3) | −0.016(5±3) | −0.09(18±11) |
| 4 | 0.09(4±5) | −0.37(4±2) | −0.33(2±3) | −0.090(3±4) | −0.160(9±8) |
| 5 | −0.16(0±5) | −0.25(7±3) | −0.21(28±17) | −0.079(5±6) | −0.10(09±12) |
| 6 | −0.28(9±8) | −0.08(8±2) | −0.05(10±15) | −0.048(5±5) | −0.026(7±7) |
| 7 | −0.29(3±6) | 0.03(6±2) | 0.06(34±14) | −0.021(7±5) | 0.023(7±6) |
| 8 | −0.23(9±8) | 0.08(9±3) | 0.10(9±2) | −0.006(2±8) | 0.044(9±8) |
| 9 | −0.18(4±8) | 0.09(2±3) | 0.09(5±5) | 0.001(7±5) | 0.04(35±12) |
| 10 | −0.10(6±6) | 0.08(2±3) | 0.06(2±3) | 0.007(0±8) | 0.030(5±5) |
| 11 | −0.03(7±5) | 0.06(25±19) | 0.02(08±20) | 0.006(2±4) | 0.01(59±10) |
| 12 | 0.02(0±7) | 0.02(0±3) | −0.02(1±3) | 0.004(7±5) | 0.00(46±13) |
| 13 | 0.0(44±10) | −0.00(6±4) | −0.05(2±5) | 0.004(7±7) | −0.00(36±14) |
| 14 | 0.0(2±2) | −0.02(3±3) | −0.06(9±7) | 0.001(9±6) | −0.01(03±16) |
| 15 | 0.0(35±17) | −0.03(9±3) | −0.06(9±8) | 0.000(5±7) | −0.01(10±11) |
| 16 | 0.0(47±14) | −0.04(1±3) | −0.06(8±7) | 0.001(6±5) | −0.01(48±12) |
| 17 | 0.04(8±8) | −0.04(4±4) | −0.02(8±3) | 0.00(07±12) | −0.01(29±13) |
| 18 | 0.04(5±4) | −0.04(0±3) | −0.02(9±3) | −0.00(02±11) | −0.00(98±18) |
| 19 | 0.04(9±6) | −0.03(8±4) | −0.02(0±3) | −0.000(6±7) | −0.009(7±9) |
| 20 | 0.0(10±10) | −0.01(9±3) | −0.00(5±3) | 0.000(6±5) | −0.00(59±15) |

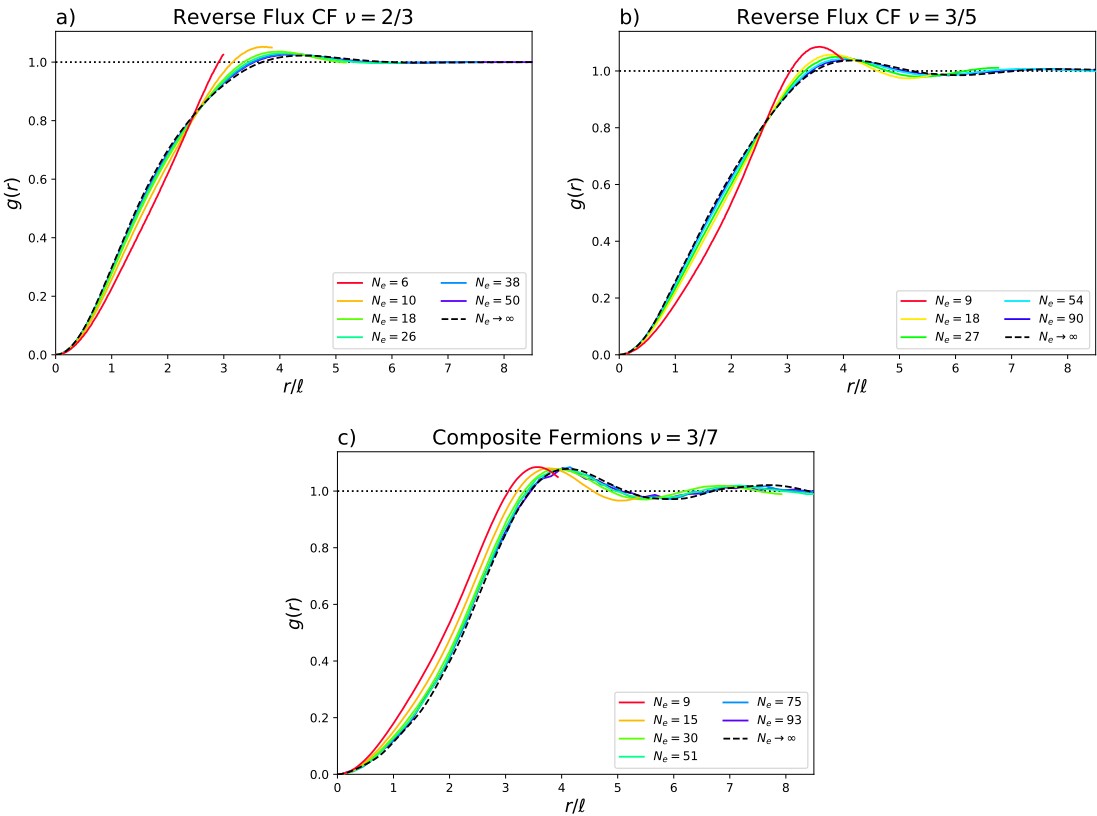

Figure 19: Pair correlation functions at finite sizes and in the thermodynamic limit for the Reverse flux composite fermion states at a) $\nu = 2/3$ b) $\nu = 3/5$ and the c) Composite Fermion state at $\nu = 3/7$.

Table 9: Expansion coefficients $c_n$ for the Moore-Read wavefunction at filling $\nu = 2 + 1/2$ and the Reverse flux modified Laughlin wavefunction at $\nu = 1/3$.

| $n$ | MR $\nu = 2+1/2$ | LCFR $\nu = 1/3$ | $n$ | MR $\nu = 2+1/2$ | LCFR $\nu = 1/3$ | $n$ | MR $\nu = 2+1/2$ | LCFR $\nu = 1/3$ |
|---|---|---|---|---|---|---|---|---|
| 1 | 1.210(9 ± 8) | 2.650(20 ± 16) | 16 | 0.01(09 ± 14) | 0.002(4 ± 5) | 31 | 0.00(00 ± 18) | 0.004(8 ± 9) |
| 2 | 0.57(5 ± 2) | 1.011(1 ± 3) | 17 | 0.00(26 ± 10) | −0.006(1 ± 6) | 32 | −0.00(1 ± 2) | 0.003(6 ± 10) |
| 3 | 0.11(02 ± 14) | −0.057(38 ± 19) | 18 | 0.00(02 ± 12) | −0.012(5 ± 7) | 33 | 0.00(37 ± 18) | 0.002(6 ± 9) |
| 4 | −0.12(23 ± 14) | −0.41(38 ± 13) | 19 | 0.00(07 ± 18) | −0.014(6 ± 5) | 34 | −0.00(12 ± 16) | 0.003(8 ± 8) |
| 5 | −0.19(04 ± 15) | −0.40(47 ± 19) | 20 | −0.00(13 ± 20) | −0.015(1 ± 4) | 35 | −0.00(3 ± 3) | 0.00(17 ± 12) |
| 6 | −0.17(57 ± 11) | −0.276(0 ± 2) | 21 | −0.00(46 ± 13) | −0.017(1 ± 6) | 36 | 0.00(2 ± 3) | 0.000(3 ± 7) |
| 7 | −0.11(99 ± 18) | −0.133(4 ± 4) | 22 | −0.00(67 ± 18) | −0.012(8 ± 4) | 37 | 0.00(1 ± 3) | 0.001(2 ± 7) |
| 8 | −0.05(6 ± 2) | −0.026(5 ± 6) | 23 | −0.00(45 ± 15) | −0.008(6 ± 4) | 38 | −0.00(1 ± 2) | 0.001(0 ± 9) |
| 9 | −0.00(99 ± 17) | 0.039(4 ± 8) | 24 | −0.00(52 ± 13) | −0.006(7 ± 5) | 39 | 0.00(1 ± 3) | 0.00(26 ± 15) |
| 10 | 0.014(6 ± 9) | 0.068(6 ± 8) | 25 | −0.00(42 ± 15) | −0.002(7 ± 6) | 40 | 0.00(5 ± 2) | 0.00(12 ± 16) |
| 11 | 0.03(21 ± 14) | 0.075(7 ± 2) | 26 | −0.00(68 ± 13) | −0.001(4 ± 5) | 41 | 0.00(3 ± 3) | −0.00(01 ± 13) |
| 12 | 0.03(76 ± 12) | 0.06(87 ± 10) | 27 | −0.00(13 ± 20) | 0.001(5 ± 4) | 42 | 0.00(3 ± 4) | −0.002(0 ± 8) |
| 13 | 0.03(65 ± 12) | 0.050(4 ± 5) | 28 | −0.00(5 ± 2) | 0.003(1 ± 4) | 43 | 0.00(2 ± 3) | −0.00(37 ± 14) |
| 14 | 0.028(5 ± 10) | 0.032(0 ± 5) | 29 | 0.00(1 ± 3) | 0.002(4 ± 7) | 44 | 0.00(3 ± 3) | −0.00(13 ± 13) |
| 15 | 0.01(72 ± 14) | 0.017(1 ± 4) | 30 | 0.00(32 ± 16) | 0.004(5 ± 5) | 45 | 0.00(4 ± 3) | −0.000(2 ± 8) |

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
