# Peer review of "Parametrization and thermodynamic scaling of pair correlation functions for the Fractional Quantum Hall Effect"

_SciPost Physics, doi:SciPost Phys. 14, 149 (2023)_

## Round 1 · Referee Report · Koyena Bose · 2023-1-4

Strengths

1. The authors have come up with a process to stably parameterize the pair-correlation functions for fractional quantum Hall states. They also propose a way to scale the pair-correlation function to the thermodynamic limit. This allows one to obtain various quantities directly in the thermodynamic limit as opposed to extrapolating results to the thermodynamic limit from evaluations on finite systems.

2. The authors have discussed and provided necessary details to explain why their parameterization of the pair correlation function is more stable than procedures deployed in the past. They have also discussed multiple techniques and showed how their parameters are convergent and robust to different choices of calculation.

3. The authors have presented $g(r)$ results for many different wave functions like Laughlin, Jain and Bonderson-Slingerland states where the thermodynamic scaling gives better results in comparison to previous calculations where computations were done for finite systems.

Weaknesses

1. The employed method still results in instabilities in the orthogonal coefficients, $G_n(\eta)$, of the parameterized pair correlation function for large $n$. A resolution of this would be required to describe the long range behaviour for more involved states like parton states.

2. More applications such as evaluating the Fermi wave vector of composite fermions (which has been measured in experiments [Phys. Rev. Lett. 113, 196801]) from the pair-correlation function could be discussed/mentioned.

Report

The study is well formulated with substantial material to support that the parameterization used for pair correlation function $g(r)$ for fractional quantum Hall states is stable in comparison to past efforts [Phys. Rev. B, 30:558–560, 1984]. The authors have also successfully applied their technique to find the pair correlation functions of some well known wave functions like Laughlin, Jain and Bonderson-Slingerland states. One of the interesting parts of this work is the scaling of $g(r)$ to the thermodynamic limit which was earlier calculated only for finite systems. This opens a path to explore various quantities in this limit which was earlier not doable , i.e. Fermi wave vector, or done for a finite system and then extrapolated to the thermodynamic limit , i.e. the evaluation of per-particle energies. It would also be interesting to see if the instability in the calculation of $G_n(\eta)$ for higher $n$ introduces considerable error when dealing with more complicated wave-functions like parton states.

Requested changes

1. For Moore-Read state with non-abelian quasi-holes [5.2; Figure: 11(c)], we see $g(r) \neq 0$ even at $r=0$. What does this signify? For a fermionic state, why is there a finite probability to find two overlapping fermions? Why does this only show up when non-abelian quasi-holes are introduced?

2. Discuss some more applications of the tools developed such as the above mentioned evaluation of the Fermi wave vector of composite fermions.

---

## Round 1 · Referee Report · Anonymous · 2023-1-25

Report

Yes the acceptance criteria of this journal are abundantly met.

---

## Editorial Decision

published